# Ketamine induces multiple individually distinct whole-brain functional connectivity signatures

Flora Moujaes[1,2]*[†], Jie Lisa Ji[1][†], Masih Rahmati[1], Joshua B Burt[3], Charles Schleifer[4], Brendan D Adkinson[5], Aleksandar Savic[6], Nicole Santamauro[1], Zailyn Tamayo[1], Caroline Diehl[7], Antonija Kolobaric[8], Morgan Flynn[9], Nathalie Rieser[2], Clara Fonteneau[1], Terry Camarro[10], Junqian Xu[11], Youngsun Cho[1,12], Grega Repovs[13], Sarah K Fineberg[1], Peter T Morgan[1,14], Erich Seifritz[2], Franz X Vollenweider[2], John H Krystal[1], John D Murray[1,3,15], Katrin H Preller[1,2][‡], Alan Anticevic[1,5][‡]

[1]Department of Psychiatry, Yale University School of Medicine, New Haven, United States; [2]Department of Psychiatry, Psychotherapy and Psychosomatics, University Hospital for Psychiatry Zurich, Zurich, Switzerland; [3]Department of Physics, Yale University, Boston, United States; [4]David Geffen School of Medicine, University of California, Los Angeles, Los Angeles, United States; [5]Interdepartmental Neuroscience Program, Yale University, New Haven, United States; [6]Department of Psychiatry, University of Zagreb, Zagreb, Croatia; [7]Department of Psychology, University of California, Los Angeles, Los Angeles, United States; [8]Center of Neuroscience, University of Pittsburgh, Pittsburgh, United States; [9]Department of Psychiatry, Vanderbilt University Medical Center, Nashville, United States; [10]Magnetic Resonance Research Center, Yale University School of Medicine, New Haven, United States; [11]Department of Radiology and Psychiatry, Baylor College of Medicine, Houston, United States; [12]Child Study Center, Yale University School of Medicine, New Haven, United States; [13]Department of Psychology, University of Ljubljana, Ljubljana, Slovenia; [14]Department of Psychiatry, Bridgeport Hospital, Bridgeport, United States; [15]Department of Psychology, Yale University, New Haven, United States

*For correspondence:
flora.moujaes@bli.uzh.ch

[†]These authors contributed equally to this work
[‡]These authors also contributed equally to this work

## Abstract

**Background:** Ketamine has emerged as one of the most promising therapies for treatment-resistant depression. However, inter-individual variability in response to ketamine is still not well understood and it is unclear how ketamine's molecular mechanisms connect to its neural and behavioral effects.
**Methods:** We conducted a single-blind placebo-controlled study, with participants blinded to their treatment condition. 40 healthy participants received acute ketamine (initial bolus 0.23 mg/kg, continuous infusion 0.58 mg/kg/hr). We quantified resting-state functional connectivity via data-driven global brain connectivity and related it to individual ketamine-induced symptom variation and cortical gene expression targets.
**Results:** We found that: (i) both the neural and behavioral effects of acute ketamine are multi-dimensional, reflecting robust inter-individual variability; (ii) ketamine's data-driven principal neural gradient effect matched somatostatin (SST) and parvalbumin (PVALB) cortical gene expression patterns in humans, while the mean effect did not; and (iii) behavioral data-driven individual symptom variation mapped onto distinct neural gradients of ketamine, which were resolvable at the single-subject level.

**Conclusions:** These results highlight the importance of considering individual behavioral and neural variation in response to ketamine. They also have implications for the development of individually precise pharmacological biomarkers for treatment selection in psychiatry.

**Funding:** This study was supported by NIH grants DP5OD012109-01 (A.A.), 1U01MH121766 (A.A.), R01MH112746 (J.D.M.), 5R01MH112189 (A.A.), 5R01MH108590 (A.A.), NIAAA grant 2P50AA012870-11 (A.A.); NSF NeuroNex grant 2015276 (J.D.M.); Brain and Behavior Research Foundation Young Investigator Award (A.A.); SFARI Pilot Award (J.D.M., A.A.); Heffter Research Institute (Grant No. 1–190420) (FXV, KHP); Swiss Neuromatrix Foundation (Grant No. 2016–0111) (FXV, KHP); Swiss National Science Foundation under the framework of Neuron Cofund (Grant No. 01EW1908) (KHP); Usona Institute (2015 – 2056) (FXV).

**Clinical trial number:** NCT03842800

---

## Editor's evaluation

This paper presents a valuable investigation of the acute effects of ketamine in healthy volunteers using rating scale-based behavioral and resting-state fMRI-based neural measures. It provides solid evidence of inter-individual variability in responses to ketamine along with associations to relevant gene expression markers (somatostatin, parvalbumin) in the brain.

---

## Introduction

Over the last two decades, ketamine has emerged as one of the most promising therapies for treatment-resistant depression (TRD) (*Krystal et al., 2019*). Robust individual differences in response to ketamine have been observed in both healthy controls and patients with TRD (*Murrough et al., 2013*; *Hack et al., 2021*). For example, a single ketamine infusion results in a response rate of around 65% in patients with TRD, while individual differences in baseline molecular effects (e.g. NMDA receptor occupancy) and brain function predict the degree to which an individual experiences specific acute ketamine-induced symptoms (*Murrough et al., 2013*; *Stone et al., 2008*; *Honey et al., 2008*). Despite these findings, an assumption persists within the fMRI pharmacology literature that the behavioral and neural effects of substances like ketamine are uniform across individuals, and that the central tendency can effectively capture such effects. In this study, we challenge this assumption and instead posit that ketamine's effects are multi-dimensional, and that these dimensions will capture individual differences in response to ketamine.

Ketamine's effects have been extensively characterised by group studies, which typically involve recruiting a sample of patients, measuring brain activity before and after ketamine infusion, and then averaging the results across the patients (i.e. relying on the power of central tendency) to ascertain ketamine's effects. At subanesthetic doses, ketamine produces transient changes in behavior, perception, and cognition that are comparable to the positive, negative, and cognitive symptoms seen in patients with psychosis-spectrum illness (*Krystal et al., 1994*). Therefore, ketamine's acute behavioral effects are typically captured using psychosis-related scales such as the Positive and Negative Syndrome Scale (PANSS), as well as cognitive tasks (*Anticevic et al., 2012*). However, the acute neural effects of subanesthetic of ketamine are less clear. The majority of studies exploring ketamine's neural alterations have used seed-based approaches, which often result in contradictory results and are fundamentally limited by the required a-priori selection of a seed region and an inability to fully capture ketamine's systems-level alterations (*Scheidegger et al., 2012*; *Niesters et al., 2012*; *Dandash et al., 2015*; *Grimm et al., 2015*; *Höflich et al., 2015*; *Khalili-Mahani et al., 2015*; *Bonhomme et al., 2016*; *Wong et al., 2016*; *Kraguljac et al., 2017*; *Liebe et al., 2018*; *Mueller et al., 2018*; *Fleming et al., 2019*; *Hack et al., 2021*; *Cole et al., 2010*; *Fitzgerald, 2012*). For instance, studies have identified both ketamine-induced increases and decreases in thalamo-cortical and hippocampal-cortical activity (*Höflich et al., 2015*; *Hack et al., 2021*; *Grimm et al., 2015*; *Khalili-Mahani et al., 2015*; *Kraguljac et al., 2017*). There is evidence that these inconsistencies may stem in part from a failure to account for individual differences in ketamine response. Ketamine has demonstrated dose-dependent individual variability in thalamic functional connectivity in healthy adults, with some individuals showing dose-dependent increases in functional connectivity, and others showing decreases (*Hack et al., 2021*).

**eLife digest** Ketamine is a widely used anesthetic as well as a popular illegal recreational drug. Recently, it has also gained attention as a potential treatment for depression, particularly in cases that don't respond to conventional therapies. However, individuals can vary in their response to ketamine. For example, the drug can alter some people's perception, such as seeing objects as larger or small than they are, while other individuals are unaffected. Although a single dose of ketamine was shown to improve depression symptoms in approximately 65% of patients, the treatment does not work for a significant portion of patients. Understanding why ketamine does not work for everyone could help to identify which patients would benefit most from the treatment.

Previous studies investigating ketamine as a treatment for depression have typically included a group of individuals given ketamine and a group given a placebo drug. Assuming people respond similarly to ketamine, the responses in each group were averaged and compared to one another. However, this averaging of results may have masked any individual differences in response to ketamine. As a result, Moujaes et al. set out to investigate whether individuals show differences in brain activity and behavior in response to ketamine.

Moujaes et al. monitored the brain activity and behavior of 40 healthy individuals that were first given a placebo drug and then ketamine. The results showed that brain activity and behavior varied significantly between individuals after ketamine administration. Genetic analysis revealed that different gene expression patterns paired with differences in ketamine response in individuals – an effect that was hidden when the results were averaged. Ketamine also caused greater differences in brain activity and behavior between individuals than other drugs, such as psychedelics, suggesting ketamine generates a particularly complex response in people.

In the future, extending these findings in healthy individuals to those with depression will be crucial for determining whether differences in response to ketamine align with how effective ketamine treatment is for an individual.

A limited number of studies have also characterised ketamine's neural systems-level functional alterations using whole-brain approaches, demonstrating that ketamine results in robust brain-wide effects. For example, ketamine has been shown to increase global brain connectivity (GBC) in the prefrontal cortex (PFC) in healthy controls specifically, and normalize the reduced PFC GBC in patients with major depressive disorder 24 hr post-ketamine (*Anticevic et al., 2015*; *Abdallah et al., 2017*; *Abdallah et al., 2018*). However, this finding failed to replicate in patients with depression 48 hr post-ketamine (*Kraus et al., 2020*), suggesting a complex relationship between ketamine's acute and delayed effects. In addition to GBC, graph theoretical approaches have demonstrated ketamine induces a shift from a cortical to a subcortically-centred brain state, particularly the basal ganglia and cerebellum (*Joules et al., 2015*). Meanwhile, a nodal predictive model found ketamine resulted in reduced connectivity within the primary cortices and the executive network, but increased connectivity between the executive network and the rest of brain (*Abdallah et al., 2021*). Finally, dynamic resting-state functional connectivity showed that ketamine decreased connectivity both within the left visual network and inter-hemispherically between the visual networks (*Spies et al., 2019*). However, these studies also fail to account for individual differences in response to ketamine.

The main limitation of relying on the power of central tendency is that meaningful differences between individual subjects may be lost through the process of averaging. One method that has been used to successfully address this and uncover individual differences in psychiatric research is a principal component analysis (PCA) (*Ji et al., 2020*; *Halai et al., 2017*). A PCA is a data-driven method that is able to uncover both group-level and individual-level differences. More specifically, a PCA allows us to test the hypothesis that ketamine's effects are multi-dimensional: if ketamine's effects are uniform then we would expect a PCA to result in one principal component on which all the participants can be mapped; however if there are systemic differences between participants, we would anticipate a PCA to generate multiple principle components. A PCA is also able to capture individual-level differences through an individual's relative positioning along the axes of the principal components (*Halai et al., 2017*).

In order to assess the relative merits of characterizing ketamine's acute effects using a uni-dimensional (i.e. mean-level) or multi-dimensional (i.e. PCA) approach, we will compare which results best relate to ketamine's hypothesized molecular mechanisms. We will focus on the indirect hypothesis, which posits that ketamine first inhibits tonic-firing GABAergic interneurons via NMDAR blockade, leading to a burst of glutamate that drives synaptic plasticity (*Gerhard et al., 2020*). Specifically, we will test the hypothesis that individual differences in ketamine's neural systems-level effects are associated with SST and PVALB GABAergic interneurons by correlating ketamine's neural effects with SST and PVALB cortical gene expression maps derived from the Allen Human Brain Atlas (AHBA). SST and PVALB genes were selected a-priori based on: (i) the large body of animal literature directly linking SST and PVALB interneurons to ketamine's acute effects (*Gerhard et al., 2020*; *Ng et al., 2018*; *Picard et al., 2019*); (ii) studies showing SST and PVALB interneurons are particularly sensitive to ketamine as a result of their NMDA receptor subunit configuration (*Paoletti et al., 2013*); (iii) previous research showing that specific cortical regions express different ratios of SST and PVALB interneurons, with SST showing higher expression levels in association relative to sensory regions, while PVALB shows the reverse pattern (*Anderson et al., 2020*; *Hawrylycz et al., 2012*); and (iv) studies demonstrating patients with depression show reduced SST or PVALB interneurons in areas such as the hippocampus and PFC (*Nowak et al., 2010*; *Zadrożna et al., 2011*; *Czeh et al., 2005*; *Czéh et al., 2015*). Given that many different genes show similar spatial patterns in the brain, with the most common pattern explaining approximately 25% of the total gene expression variance (*Burt et al., 2018*), it is important to note that by relating ketamine's neural effects to SST and PVALB cortical gene expression patterns we do not make any claims regarding gene specificity. Instead, we use the established relationship between ketamine and SST/PVALB interneurons to evaluate the relative merits of assessing ketamine's acute effects using a multi-dimensional vs. uni-dimensional approach.

The current study aims to investigate the behavioral and neural effects of acute ketamine administration in 40 healthy controls using a whole-brain data-driven multi-dimensional (PCA) approach. We hypothesize that ketamine will result in multiple neural and behavioral axes of variation, reflecting inter-individual variability in response to ketamine. We further hypothesize that using a multi-dimensional vs. uni-dimensional approach to capture ketamine's acute effects will allow us to more closely relate ketamine's effects to its hypothesized molecular mechanisms, specifically SST and PVALB cortical gene expression maps. In addition, we investigate the extent to which multi-dimensionality is a property specific to ketamine by directly comparing ketamine to two other psychoactive substances that have also shown preliminary clinical efficacy for the treatment of depression, but have very different pharmacological profiles: (i) psilocybin, a preferential 5-HT2A and 5-HT1A agonist; and (ii) LSD, which stimulates a wide range of serotonin and dopamine receptors (*Gasser et al., 2014*; *Carhart-Harris et al., 2016*; *Pokorny et al., 2016*; *De Gregorio et al., 2016*; *Halberstadt and Geyer, 2011*; *Passie et al., 2008*). Finally, we compare the principal behavioral axes to established psychosis literature subscales (e.g. three-factor and five-factor PANSS models), and assess the extent to which the principal behavioral axes captures novel neural variation and individual differences in molecular mechanisms.

## Materials and methods

**Key resources table**

| Reagent type (species) or resource | Designation | Source or reference | Identifiers | Additional information |
|---|---|---|---|---|
| Software, algorithm | QuNex | QuNex, *Ji et al., 2023* | | |
| Software, algorithm | HCP MPP | HCP MPP, *Glasser et al., 2013* | | |
| Software, algorithm | N-BRIDGE | N-BRIDGE, *Ji et al., 2020* | | |
| Software, algorithm | CAB-NP | CAB-NP, *Ji et al., 2019* | | |
| Software, algorithm | R | R | RRID:SCR_001905 | |
| Software, algorithm | FSL | FSL | RRID:SCR_002823 | |

### Ketamine study participants

Healthy participants were recruited from the New Haven area via flyers and online ads. In order to be eligible to receive ketamine, participants were required to meet the following set of criteria, as

determined by a detailed telephone interview and an in-person clinical assessment: (i) Age 21–60; (ii) IQ >70 as measured via Wide Range Achievement Test (WRAT-3) and the Wechsler Adult Intelligence Scale (WAIS-III); (iii) intact or corrected-to-normal vision; (iv) weight <300 lbs.; (v) MR safe (free of metallic objects and absence of claustrophobia); (vi) no serious medical or physical conditions, as confirmed by a self-report, electrocardiogram, blood work, and physical examination by a licensed physician; (vii) no lifetime neurological or psychiatric diagnoses; (viii) minimal alcohol intake and no use of psychoactive drugs or history of abuse/dependence, confirmed by interview and urinalysis; (ix) no first-degree relatives with DSM Axis-I diagnoses or alcohol/substance abuse history; (x) no known sensitivity to ketamine or heparin; (xi) no donation of blood in excess of 500 ml within 2 months of participation. Eligible subjects provided informed consent approved by Yale University Institutional Review Board. Forty participants took part in the study. The sample size was determined ahead of study initiation based on prior studies *Anticevic et al., 2011*; *Driesen et al., 2008*; *Johnson et al., 2006*; *Driesen et al., 2013*; *Anticevic et al., 2015*, which would achieve statistical power of >86% for a estimated medium effect size (Cohen's d of 0.5). Demographic details can be found in *Supplementary file 2*.

## Ketamine study design

The protocol was approved by the Yale Human Investigations Committee (ClinicalTrials.gov Identifier: NCT03842800). All subjects provided written informed consent. The study employed a single-blind within-subjects design, with participants blinded to their treatment condition. Placebo was administered during the first neuroimaging scan session, and ketamine (initial bolus 0.23 mg/kg, continuous infusion 0.58 mg/kg/hr) during the second, as residual ketamine effects ruled out counterbalancing of the order. Participants were not informed about the order in which they would receive saline and ketamine, however they were able to correctly identify the ketamine infusion 100% of the time. Cognitive effects were measured using a Spatial Working Memory task completed in the scanner. Subjective effects were measured before and after the scan (180 min post drug administration) using the following scales: (1) Positive and Negative Syndrome Scale (PANSS), (2) Clinician Administered Dissociative States Scale (CADSS), and (3) Beck's Depression Inventory (BDI).

## Ketamine infusion protocol

Participants were instructed to fast for 12 hr prior to the scan and avoid any alcohol or medications 72 hr prior to the scan. As a precaution, blood alcohol content was assessed with an electronic

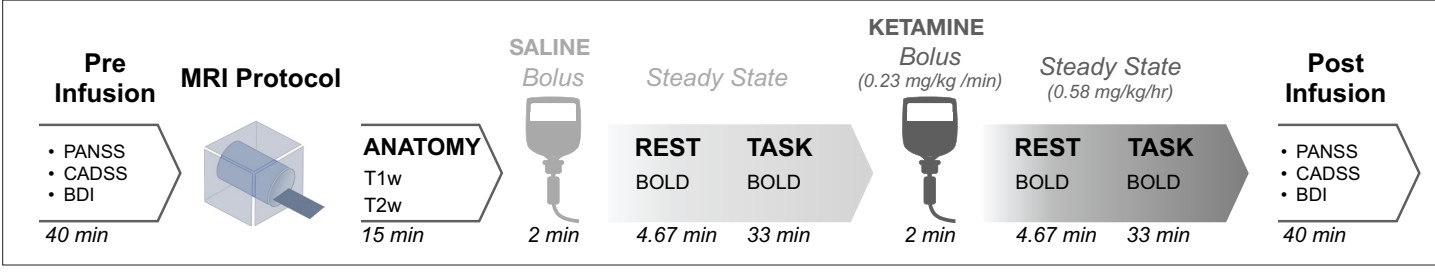

**Figure 1.** Neuroimaging protocol. The study employed a single-blind within-subjects design, with participants blinded to their treatment condition. Forty participants received an IV cannulation in each forearm: one IV for the placebo/ketamine infusion and one for blood draws during the scan. Placebo was administered during the first neuroimaging can session, and ketamine (initial bolus 0.23 mg/kg, continuous infusion 0.58 mg/kg/hr) during the second because residual ketamine effects ruled out counterbalancing of the order. Blood was drawn immediately after the resting-state run. Cognitive effects were measured using a Spatial Working Memory task completed in the scanner. Subjective effects were measured 180 min post drug administration using the following scales: (1) Positive and Negative Syndrome Scale (PANSS), (2) Clinician Administered Dissociative States Scale (CADSS), and (3) Beck's Depression Inventory (BDI).

The online version of this article includes the following figure supplement(s) for figure 1:

**Figure supplement 1.** Mean effect of ketamine on global brain connectivity without global signal regression.

**Figure supplement 2.** Multi-dimensional neural effect of acute ketamine administration without global signal regression.

**Figure supplement 3.** Comparison of Delta GBC Principal Component Analysis (PCA) with vs. without Global Signal Regression(GSR).

**Figure supplement 4.** Mean effect of ketamine on GBC.

**Figure supplement 5.** Effect of sex and age on neural and behavioral PC scores.

breathalyzer on the morning of the infusion. Prior to entering the scanner, subjects received IV cannulation in each forearm: one IV for the placebo/ketamine infusion and one for blood draws during the scan. In the first scan session, an initial bolus of saline was administered over two minutes prior to collecting BOLD images, followed by a continuous maintenance infusion of saline throughout the remainder of the scan. In the second scan session, an initial bolus of racemic ketamine (0.23 mg/kg) diluted in saline solution was delivered just prior to data acquisition, followed by a continuous infusion of racemic ketamine (0.58 mg/kg/hr) throughout the remainder of the scan. Blood was drawn immediately after the resting-state run, and subsequent gas chromatography-mass spectrometry determined that the group mean serum ketamine concentration at the end of this run was 470 nmol/L. The scan was aborted and the participant removed for evaluation if, at any point, they expressed discomfort related to administration of the drug or placebo, or if they became unresponsive or showed any indications of an adverse reaction. Diazepam was available for anxiety in the event of a particularly negative response, but was never needed. Full neuroimaging protocol details can be found in (*Figure 1*).

## Ketamine neuroimaging data acquisition

Neural data were collected using a Siemens 3T scanner with with a 32 channel head coil at the Yale Center for Biomedical Imaging. Imaging acquisition parameters were aligned with those of the Human Connectome Project (HCP) (*Van Essen et al., 2013*). High-resolution T1w and T2w structural images were acquired in 224 AC-PC aligned slices, 0.8 mm isotropic voxels. T1w images were collected with a magnetization-prepared rapid gradient-echo (MP-RAGE) pulse sequence (time repetition (TR)=2400ms, time echo (TE)=2.07ms, flip angle = 8°, field of view = 256 x 256 mm). T2w images were collected with a SCP pulse sequence (TR = 3200ms, TE = 564ms, flip angle = T2 var, field of view = 256 x 256 mm). Resting-state BOLD images were collected with a multi-band accelerated fast gradient-echo, echo-planar sequence (acceleration factor = 6, TR = 700ms, TE = 31.0ms, flip angle = 55°, field of view = 210 x 210 mm, matrix = 84 x 84, bandwidth = 2290 Hz); 54 interleaved axial slices aligned to the anterior-posterior commissure (AC-PC) with 2.5 mm isotropic voxels. A total of 400 volumes were acquired per resting state scan resulting in a scan duration of 4.67 min. Additionally, a pair of reverse phase-encoded spin-echo field maps (anterior-to-posterior and posterior-to-anterior) were acquired (voxel size = 2.5 mm isotropic, TR = 7220ms, TE = 73ms, flip angle = 90°, field of view = 210 × 210 mm, bandwidth = 2290 Hz).

## LSD and psilocybin neuroimaging data acquisition

To assess the extent to which inter-indiviudal variability is specific to ketamine, we compared the neural effects of ketamine to the neural effects of lysergic acid diethylamide (LSD) and psilocybin. For the LSD and psilocybin analyses, we utilized data from two independent pharmacological neuroimaging datasets (*Preller et al., 2018*; *Preller et al., 2020*). Methods for the LSD neuroimaging study (N=24) are described in detail in prior publications (*Preller et al., 2018*). The use of LSD in humans was authorized by the Swiss Federal Office of Public Health, Bern, Switzerland. The study protocol was approved by the Cantonal Ethics Committee of Zurich (KEK-ZH_No: 2014_0496). The study employed a fully double-blind, randomized, within-subject cross-over design with three conditions: (1) placebo +placebo condition: placebo (179 mg Mannitol and Aerosil 1 mg po) after pretreatment with placebo (179 mg Mannitol and Aerosil 1 mg po); (2) placebo +LSD condition: LSD (100 µg po) after pretreatment with placebo (179 mg Mannitol and Aerosil 1 mg po), or (3) Ketanserin +LSD condition: LSD (100 µg po) after pretreatment with the 5-HT2A antagonist Ketanserin (40 mg po). Data were collected for all subjects in a randomized counterbalanced order at three different sessions each two weeks apart. For all conditions, the first substance was administered 60 min before the second substance, and the first neural scan was conducted 75 min after the second administration, with a second scan conducted at 300 min post-administration. In the present study, only data from the (1) placebo +placebo and (2) placebo +LSD conditions were evaluated. Methods for the psilocybin neuroimaging study (N=23) are described in detail in prior publications (*Preller et al., 2020*). The use of psilocybin in humans was authorized by the Swiss Federal Office of Public Health, Bern, Switzerland. The study was registered at ClinicalTrials.gov (NCT03736980). The study employed a fully double-blind, randomized, within-subject cross-over design. Participants at two different occasions 2 weeks apart received either placebo (179 mg mannitol and colloidal silicon dioxide [Aerosil; Evonik Resource Efficiency GmbH, Essen, Germany] 1 mg orally; placebo condition) or psilocybin

(0.2 mg/kg orally; psilocybin condition). The resting-state scan was conducted at three time points between administration and peak effects: 20, 40, and 70 min after treatment administration. In the present study, only data from the psilocybin and placebo neural scans conducted at 70 min were evaluated.

## Neuroimaging data preprocessing

Neuroimaging data was preprocessed with the Human Connectome Project (HCP) minimal preprocessing pipeline (*Glasser et al., 2013*) using the Quantitative Neuroimaging Environment & Toolbox (QuNex; *Ji et al., 2023*). A summary of the HCP pipelines is as follows. First T1/2-weighted structural images were corrected for bias-field distortions and then warped to the standard Montreal Neurological Institute-152 (MNI-152) brain template in a single step, through a combination of linear and non-linear transformations via the FMRIB Software Library (FSL) linear image registration tool (FLIRT) and non-linear image registration tool (FNIRT; *Jenkinson et al., 2002*). Next, FreeSurfer's recon-all pipeline was used to segment brain-wide gray and white matter to produce individual cortical and subcortical anatomical segmentations (*Reuter et al., 2012*). Cortical surface models were generated for pial and white matter boundaries as well as segmentation masks for each subcortical gray matter voxel. Using the pial and white matter surface boundaries, a 'cortical ribbon' was defined along with corresponding subcortical voxels, which were combined to generate the neural file in the Connectivity Informatics Technology Initiative (CIFTI) volume/surface 'grayordinate' space for each individual subject (*Glasser et al., 2013*). BOLD data were motion-corrected by aligning to the middle frame of every run via FLIRT in the initial NIFTI volume space. In turn, a brain-mask was applied to exclude signal from non-brain tissue. Next, cortical BOLD data were converted to the CIFTI gray matter matrix by sampling from the anatomically-defined gray matter cortical ribbon and subsequently aligned to the HCP atlas using surface-based nonlinear deformation (*Glasser et al., 2013*). Subcortical voxels were aligned to the MNI-152 atlas using whole-brain non-linear registration and then the Freesurfer-defined subcortical segmentation applied to isolate the subcortical grayordinate portion of the CIFTI space.

After the HCP minimal preprocessing pipelines, movement scrubbing was performed (*Power et al., 2013*). Bad frames with possible movement-induced artifactual fluctuations in intensity were identified if they met at least one of the following two criteria (*Anticevic et al., 2012*). First, framewise displacement (FD) was computed by summing the displacement across all six rigid body movement correction parameters. Frames in which FD exceeded 0.5 mm were flagged. Secondly, normalized root mean square (RMS) was calculated by taking the root mean square of differences in intensity between the current and preceding frame across all voxels and dividing it by the mean intensity. Frames in which RMS exceeded 1.6 x the median across the scans (calculated separately for each participant) were also flagged. The flagged frames, as well as the the frame immediately preceding and immediately following any flagged frames, were discarded from further analyses. Subjects with more than 50% flagged frames were excluded completely.

Next, a high-pass filter (threshold 0.008 Hz) was applied to the BOLD data to remove low frequency signals due to scanner drift. QuNex was used to calculate the average variation in BOLD signal in the ventricles, deep white matter, and across the whole gray matter ('global signal'), as well as movement parameters. These signals, as well as their first derivatives to account for delayed effects, were then regressed out of the gray matter BOLD time series as nuisance variables (as any change in the BOLD signal due to these variables are not likely to reflect neural activity) (*Power et al., 2018*). It should be noted that using global signal regression (GSR) to remove spatially persistent artifact is controversial in neuroimaging (*Power et al., 2017*; *Yang et al., 2017*), but it remains a current field-wide standard (although see other recent and emerging approaches at *Glasser et al., 2018*; *Aquino et al., 2020*). In order to address this controversy, we also include the mean effect and neural PCA results without GSR (*Figure 1—figure supplement 1*, *Figure 1—figure supplement 2*). In addition, when we compared the results of the neural PCA performed with and without GSR, we found that PC1 Δ GBC with GSR was significantly positively correlated with the PC2 Δ GBC without GSR ($r$=0.54, p<0.001) (*Figure 1—figure supplement 3E*). In contrast, PC1 Δ (ketamine - placebo) GBC with GSR was not significantly correlated with the PC1 Δ GBC without GSR ($r$=–0.04, p=0.28; *Figure 1—figure supplement 3D*). This indicates that when a PCA is computed on results without GSR, the first PC captures a substantial amount of the global signal that is removed when regressing out GSR.

## Neural data reduction via functional brain-wide parcellation

All neural data analysed in this study is resting-state data. We examined the neural effects of ketamine at multiple levels of analysis: dense (91,281 grayordinates); parcel (718 parcels); network (12 networks); and subcortex (5 subcortical structures). Psilocybin and LSD were analysed at the parcel-level only. Both the parcel-level and network-level data were acquired using a functionally derived whole-brain parcellation via the recently validated Cole-Anticevic Brain Network Parcellation (CAB-NP) atlas (*Ji et al., 2019*; *Glasser et al., 2016*). The subcortex-level data was acquired by parcellating the neural data using Freesurfer's anatomically defined subcortical structures. All data was parcellated prior to running GBC, as this was shown to improve the signal-to-noise ratio (*Figure 2—figure supplement 1*). While we report the mean effect of ketamine at grayordinate and parcel-level (*Figure 1—figure supplement 4*), further analysis was conducted at the parcel-level as this gave the best trade-off between the sample size needed to resolve multivariate neurobehavioral solutions and the size of the feature space (*Ji et al., 2020*).

## Global brain connectivity

GBC was calculated on the ketamine, psilocybin, and LSD datasets. Following preprocessing, the resting-state functional connectivity (FC) matrix was calculated for each participant by computing the Pearson's correlation between every grayordinate in the brain with all other grayordinates. A Fisher's r-to-Z transform was then applied. GBC was calculate by computing every grayordinate's mean FC strength with all other grayordinates (i.e. the mean, per row, across all columns of the FC matrix). Thus, this calculation yielded a GBC map for each subject where each grayordinates value represents the mean connectivity of that grayordinate with all other grayordinates in the brain. GBC is a data-driven summary measure of connectedness that is unbiased with regards to the location of a possible alteration in connectivity (*Cole et al., 2016*) and is therefore a principled way for reducing the number of neural functional connectivity features while assessing neural variation across the entire brain. GBC was calculated as:

$$GBC(x) = \frac{1}{N} \sum_{y=1}^{N} r_{xy}$$

where $GBC(x)$ denotes the GBC value at grayordinate $x$; $N$ denotes the total number of grayordinates; $\sum_{y=1}^{N}$ denotes the sum from $y = 1$ to $y = N$; and where $r_{xy}$ denotes the correlation between the time-series of grayordinates $x$ and $y$. For parcel-level and network-level maps, as outlined in the previous section we first computed the mean BOLD signal within each parcel/network for each participant, and then computed the pairwise FC between all parcels/networks. Finally, to obtain the parcellated GBC metric we computed the mean FC for each parcel/network.

## Principal component analysis of neural data

GBC was calculated on the ketamine, psilocybin, and LSD datasets. For each dataset the input for the PCA of the neural data was the parcel-level Δ (substance - placebo) GBC maps for each subject. The PCA was computed using the 718 parcels across all participants. Significance of the neural PCA solution was assessed via permutation testing (1000 random shuffles of parcels within subject). Finally, we also assessed whether differences in sex and age were related to neural PC scores (*Figure 1—figure supplement 5A–O*).

## Effective dimensionality

Effective dimensionality was calculated on the ketamine, psilocybin, and LSD datasets to compare the dimensionality of the neural effects of different pharmacological substances. We used the participation ratio (PR), calculated as:

$$PR = (\Sigma_i \bar{\lambda}_i^2)^{-1}$$

where $\{\lambda_i\}$ is the *ith* eigenvalue of the covariance matrix, and $\bar{\lambda}_i = \lambda / \Sigma_i \lambda_i$ (*Ehrlich and Murray, 2021*; *Gao et al., 2017*). Larger values indicate a more complex higher dimensional dataset, while smaller values indicate a less complex lower dimensional dataset. Input for the calculation for the LSD condition were the Δ (LSD - placebo) GBC maps for each participant, input for the psilocybin condition

were the Δ (psilocybin - placebo) GBC maps for each participant, and input for the ketamine condition were the Δ (ketamine - placebo) GBC maps for each participant. As sample size is relevant when calculating effective dimensionality, we first re-sampled the data ensuring sample size in each of the pharmacological conditions was always 22. We used either re-sampling or jackknifing to build a distribution of effective dimensionality values for each pharmacological condition. For LSD (N=24), we selected all 276 possible combinations of 22 participants and calculated effective dimensionality on each of the 276 subsamples to build a distribution. For psilocybin (N=23), we used jackknifing to produce 22 subsamples of 22 participants and calculated effective dimensionality on each of the 22 subsamples to build a distribution. For ketamine (N=40) we randomly selected 100 subsamples from the 113380261800 possible combinations of 22 participants and calculated effective dimensionality on each of the 100 subsamples to build a distribution. To compare the pharmacological conditions we then ran a one-way ANOVA comparing effective dimensionality distributions across LSD, psilocybin, and ketamine. Finally, we ran a series of post-hoc t-tests with Bonferroni correction.

## Neural gene expression mapping

Neural gene expression mapping was calculated on the ketamine dataset only. Methods for the gene mapping analyses in this study are described in detail in prior publications (*Ji et al., 2020*; *Burt et al., 2018*). To relate ketamine-specific neuroimaging effects to the cortical topography of gene expression for candidate receptors, we used cortical gene expression data from the publicly available Allen Human Brain Atlas (AHBA, RRID:SCR-007416), mapped to cortex (*Burt et al., 2018*). Specifically, the AHBA quantified expression levels across 20,737 genes obtained from six postmortem human brains using DNA microarray probes sampled from hundreds of neuroanatomical loci. We mapped gene expression on to 180 symmetrized cortical parcels from the HCP atlas *Glasser et al., 2016* in line with recently published methods (*Ji et al., 2020*). This yielded a group-level map for each gene where the value in each parcel reflected the average expression level of that gene in the AHBA dataset. We selected two interneuron marker genes (somatostatin [SST] and parvalbumin [PVALB]) (*Higgins et al., 2020*). As in prior works, we first excluded any gene expression maps where the cortical differential stability value was between 0 and +/-0.1 (*Burt et al., 2018*). We then correlated the gene expression maps for each of the selected genes with our target map. As the gene expression maps are restricted to the cortex, the correlations were run on cortex only. To assess the significance of each correlation, we used the following approach, which is outlined in more detail in *Burt et al., 2020* and *Figure 3—figure supplement 2*. We first generated 100,000 surrogate maps, whose spatial autocorrelation was matched to the spatial autocorrelation of the target map. The surrogate maps were generated separately for the left and right hemispheres and then merged together to produce 100,000 whole-brain cortical surrogate maps. For each selected gene, we correlated the gene expression map with each of the 100,000 surrogate maps to get a distribution of 100,000 simulated r values. We then used this distribution of simulated r values to calculate the significance of the correlation between the gene expression map and the target map. All p-values were FDR corrected.

## Principal component analysis of behavioral measures

The PCA of behavioral measures was calculated on the ketamine dataset only. Cognition was analyzed using a spatial working memory paradigm, which resulted in a single cognition score. Subjective effects were analyzed using the Positive and Negative Syndrome Scale (PANSS), an assessment of psychosis symptom severity (*Kay et al., 1987*). The full PANSS battery is conventionally divided into three subscales: Positive symptom scale (7 items), Negative symptom scale (7 items), and General Psychopathology symptom scale (16 items). In total, this yields 31 symptom variables per participant. Two participants had missing values for PANSS, so the mean was imposed. Behavioral response to ketamine assessed via PANSS showed sufficient range to justify further analysis, as 35/40 participants showed a significant difference in PANSS following ketamine, while outliers were limited (with the exception of items in which the interquartile range was 0) (*Supplementary files 3-4*). We did not include CADSS in the model as doing so resulted in the total amount of variance explained by the PCA going down from 41.1% to 29.9% (*Figure 4—figure supplement 2*). For comparison, we ran a PCA on (i) PANSS, CADSS, & cognition (*Figure 4—figure supplement 2*) and (ii) CADSS, & cognition (*Figure 4—figure supplement 3*), and directly compared the different PCA versions by correlating the resulting neuro-behavioral maps (*Figure 5—figure supplement 1*).

The PCA of behavioral data was computed using the 31 symptom variables across all N=40 participants. Variables were first scaled to have unit variance across participants before running the PCA. Significance of the derived principal components (PCs) was computed via permutation testing. For each permutation, participant order was randomly shuffled for each symptom variable before re-computing the PCA. This permutation was repeated 5000 times to establish the null model. PCs which accounted for a proportion of variance that exceeded chance (p<0.05 across all 5000 permutations) were retained for further analysis. Finally, we also assessed whether differences in sex and age were related to behavioral PC scores (*Figure 1—figure supplement 5P–U*).

## Mass univariate symptom-neural mapping

The mass univariate symptom-neural mapping was calculated on the ketamine dataset only. Methods for the mass univariate symptom-neural mapping are described in detail in prior publications (*Ji et al., 2020*). Behavioral scores were quantified in relation to individual GBC variation at the parcel-level via a mass univariate regression procedure. The resulting maps of regression coefficients reflects the strength of the relationship between participants' behavioral score and Δ GBC at every neural location (718 parcels), across all 40 participants. The greater the magnitude of the coefficient for a given location, the stronger the statistical relationship between Δ GBC and the behavioral variation across participants. Significance of the maps was assessed via nonparametric permutation testing, 5000 random shuffles with TFCE (*Smith and Nichols, 2009*) type-I error-protection computed via the Permutation Analysis of Linear Models program (*Winkler et al., 2014*).

## Results

### Multidimensional neural effect of acute ketamine administration

To explore whether the neural effects of acute ketamine are multidimensional (i.e. there is inter-individual variability in response to ketamine) or uni-dimensional (i.e. ketamine effects individuals in a similar manner), we ran a PCA. The PCA was computed by: (i) parcelling the neural data into 718 functionally-defined parcels (*Figure 2—figure supplement 1*); (ii) computing GBC for each parcel; (iii) calculating the Δ (ketamine - placebo) GBC map for each participant; and (iv) performing a PCA on the Δ GBC neural features (718 parcels) across participants (N=40). For comparison, a PCA was also performed separately on the placebo and ketamine GBC maps (*Figure 2—figure supplement 2*).

The PCA performed on the Δ GBC maps revealed ketamine results in five significant bi-directional PCs (*Figure 2A*, *Figure 2—figure supplement 4*) that together capture 42.1% of the total variance. In the main text, we focus on PC1-2, which each capture a much greater proportion of the variance (>10%) than PC3-5 (approximately 5%). Results for PC3-5 are reported in the supplement (*Figure 2—figure supplement 3*, *Figure 2—figure supplement 4*, *Figure 3—figure supplement 1*). To explore whether there is network specificity in the multi-dimensional ketamine effect, we also grouped parcels into sensory or association networks (*Figure 2—figure supplement 4*). This revealed PC1-2 Δ GBC successfully differentiate between association and sensory networks (*Figure 2D, E and G & H*). It is important to note that as PC1-2 Δ GBC are bi-directional axes of variation, some individuals exhibit increased GBC in association networks and decreased GBC in sensory networks, while others show the opposite pattern (*Figure 2—figure supplement 3A–D*).

As a control, we also examined whether ketamine's multi-dimensional neural effects are specific to ketamine or a general property of pharmaco-neuroimaging. We compared the acute effect of ketamine to two drugs that also have anti-depressive effects, psilocybin and LSD, by: (i) calculating Δ GBC parcelled neural maps for each participant; and (ii) computing effective dimensionality for each substance using the Δ GBC maps as input (see Materials and methods). Ketamine resulted in significantly higher dimensionality compared to LSD and psilocybin, while there was no difference between LSD and psilocybin (*Figure 2C*). Furthermore, we found no evidence differences in effective dimensionality were driven by motion (*Figure 2—figure supplement 5*). Finally, for comparison, we also compared ketamine, psilocybin, and LSD using a PCA (*Figure 2—figure supplements 6 and 7*).

We then computed the mean map by averaging the mean Δ GBC map across participants (*Figure 2J*). Comparing the PCA and mean approaches revealed that there is variation within the neural response to ketamine that is not sufficiently captured using the mean: while mean Δ GBC is moderately correlated with PC1 Δ GBC, it is only weakly correlated with PC2-5 Δ GBC (*Figure 2B*).

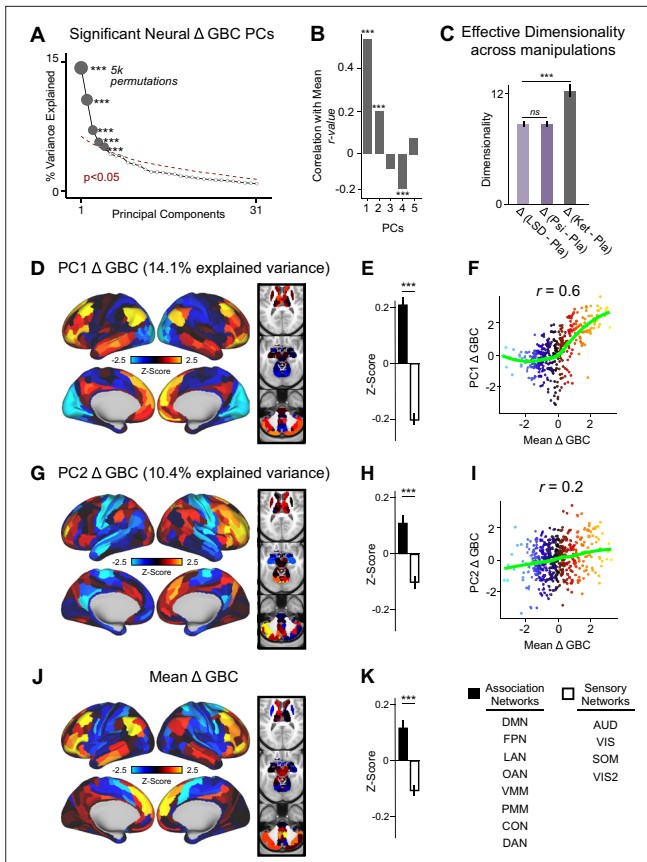

**Figure 2.** Multidimensional neural effect of acute ketamine administration. (**A**) Results of PCA performed on Δ GBC neural features (718 whole-brain parcel GBC) across all subjects (N=40). Screeplot showing the % variance explained by the first 31 (out of 39) Δ GBC PCs. The first 5 Δ GBC PCs (dark gray) were determined to be significant using a permutation test (p<0.05, 5000 permutations). The size of each dark gray point is proportionate to the variance explained. Together, these 5 Δ GBC PCs capture 42.1% of the total variance in neural GBC in the sample. (**B**) Bar plot showing the correlation between each of the five significant Δ GBC PCs maps and the mean Δ GBC map. r values for PC1, PC2, and PC4 Δ GBC are significant (***=p < 0.001, Bonferroni corrected). (**C**) Bar plot showing the effective dimensionality across three pharmacological conditions: Δ LSD (LSD - placebo), Δ psilocybin (psilocybin - placebo), and Δ ketamine (ketamine - placebo). (Light purple = LSD, dark purple = psilocybin, dark gray = ketamine). There was a statistically significant difference between groups as determined by a one-way ANOVA (F(2,60)=564, p<0.001). Post hoc tests revealed that dimensionality was significantly lower in the LSD (8.7+/-0.3, p<0.001) and the psilocybin (8.6+/-0.3, p<0.001) conditions compared to the ketamine (12.8±0.7) condition. There was no statistically significant difference in dimensionality between LSD and psilocybin (p=0.6). Error bars = standard deviation. (**D**) Unthresholded PC1 Δ GBC Z-score map (Z-scores computed across 718 parcels). PC1 Δ GBC explains 14.1% of all variance. Red/orange areas = parcels that have a high positive loading score onto PC1, blue areas = parcels that have a high negative loading score onto PC1. (**F**) Scatter plot showing the relationship across parcels between mean Δ GBC and PC1 Δ GBC maps (r=0.56, p<0.001, n=718). (**G**) Unthresholded PC2 Δ GBC Z-score map (Z-scores computed across 718 parcels). PC2 Δ GBC explains 10.4% of all variance. Red/orange areas = parcels that have a high positive loading score onto PC2, blue areas = parcels that have a high negative loading score onto PC2. (**H**) Bar plot showing the PC2 Δ GBC Z score across association (black) and sensory (white) network parcels. (**I**) Scatter plot showing the relationship across parcels between mean Δ GBC and PC2 Δ GBC maps (r=0.23, p<0.001, n=718). Green line indicates neither the positive or negative values are highly correlated. (**J**) Unthresholded mean Δ (ketamine - placebo) GBC Z-score map (Z-scores computed across 718 parcels). Red/orange areas = parcels where participants exhibited stronger GBC in the ketamine vs. placebo condition, blue areas = parcels where participants exhibited reduced GBC in the ketamine vs. placebo condition. (**K**) Bar plot showing the mean Δ GBC Z-score across association (black) and sensory (white) network parcels. (DMN = default mode, FPN = frontoparietal, LAN = language, OAN = orbito-affective, VMM = ventro multimodal, PMM

*Figure 2 continued on next page*

*Figure 2 continued*

= posterior multimodal, CON = cingulo-opercular, DAN = dorsal attention, AUD = auditory, VIS = primary visual, SOM = somatomotor, VIS2=secondary visual).

The online version of this article includes the following figure supplement(s) for figure 2:

**Figure supplement 1.** Parcellating before running GBC results in an improved signal to noise ratio.

**Figure supplement 2.** Principal component analysis (PCA) on the neural features from the placebo, ketamine, and Δ (ketamine- placebo) neural GBC maps.

**Figure supplement 3.** Individual variation in PC1-5 Δ GBC and mean Δ GBC.

**Figure supplement 4.** Network breakdown of PC1-5 Δ GBC and mean Δ GBC.

**Figure supplement 5.** Investigating the relationship between motion, GBC variance, and effective dimensionality across threedatasets: ketamine, LSD, and psilocybin.

**Figure supplement 6.** Principal Component Analysis of LSD and psilocybin Δ GBC.

**Figure supplement 7.** A comparison of principal component analysis results across the following datasets: placebo, ketamine, Δ, Δ psilocybin, and Δ LSD.

Furthermore, PC1 Δ GBC captures more variation than the mean Δ GBC (*Figure 1—figure supplement 3B, L*), and the two maps do not have a linear relationship: though the positive values are highly correlated, the negative values show no relationship (*Figure 2F*).

Overall, this suggests that ketamine results in robust inter-individual variability that is not fully captured using a mean-based approach, and that this variability may be higher than in substances such as psilocybin and LSD.

## Ketamine's data-driven principal neural gradient is associated with SST and PVALB cortical gene expression patterns, while the mean effect is not

We hypothesized that analysing ketamine's neural effects using a multi-dimensional approach rather than a mean-based approach would result in a stronger association with SST/PVALB cortical gene expression profiles. To test this, we evaluated the relationship between PC1-5 Δ GBC and SST/PVALB gene expression maps using the method outlined in (*Figure 3*, *Figure 3—figure supplement 2*). We found that as hypothesized, the multi-dimensional approach was more strongly associated with SST/PVALB cortical gene expression profiles. For example, PC1 Δ GBC was significantly correlated with both the SST ($r$=0.47, p<0.001, 100,000 permutations) and PVALB ($r$=−0.22, p=0.025, 100,000 permutations) cortical gene expression maps, well beyond what is expected by chance alone (*Figure 3E*). In contrast, no significant correlations were found between mean Δ GBC and SST/PVALB cortical gene expression maps (*Figure 3H*). Furthermore, PC3-5 Δ GBC also captured the association between ketamine-induced changes in GBC and SST/PVALB cortical gene expression maps (*Figure 3—figure supplement 1J, M and P*).

## Multidimensional behavioral effect of acute ketamine administration

To assess whether the behavioral effects of acute ketamine are also multi-dimensional, we performed a PCA on the 31 Δ (ketamine - placebo) behavioral measures collected across all 40 participants. The behavioral measures were collected using: (i) an objective measure of cognition (spatial working memory task) collected during the scan session; and (ii) a subjective measure of positive and negative symptom-like effects (PANSS) collected retrospectively 180 min after drug administration. The marked collinearity between behavioral measures across the existing PANSS subscales indicates a dimensionality-reduced solution such as a PCA may warranted in order to capture meaningful variation in participants' behavioral response (*Figure 4A–B*).

The PCA showed ketamine results in two significant bi-directional axes of behavioral alteration, that together capture 41.4% of all variance (*Figure 4C*). The loading configuration of the behavioral measures that form each PC shows that behavioral PC1 indexes variation particularly in the regards to: (i) negative-symptom items such as blunted affect, emotional withdrawal, social withdrawal, abstract thought, and lack of spontaneity; (ii) the positive-symptom item conceptual disorganisation; and (iii) general-symptom items such as tension, motor retardation, poor attention, and lack of insight and

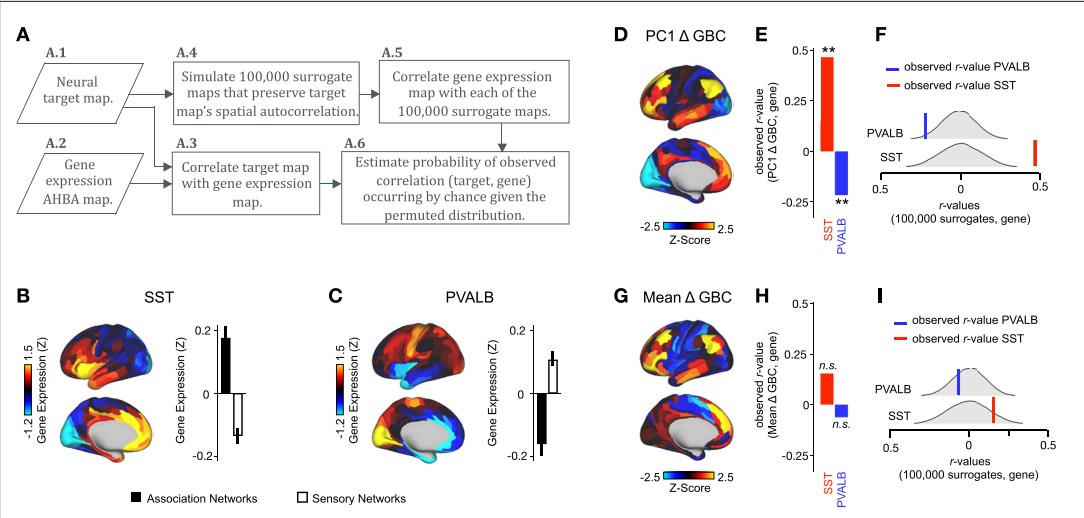

**Figure 3.** PC1 Δ GBC map is associated with SST and PVALB cortical gene expression patterns. (**A**) Gene analysis workflow. (**A.1**) Selection of neural target map. (**A.2**) Cortical gene expression map was obtained using GEMINI-DOT (***Burt et al., 2018***). Specifically, an AHBA gene expression map was obtained using DNA microarrays from six postmortem human brains, capturing gene expression topography across cortical areas. These expression patterns were then mapped onto the cortical surface models derived from the AHBA subjects' anatomical scans and aligned with the Human Connectome Project (HCP) atlas, described in prior work and Methods (***Burt et al., 2018***). (**A.3**) Correlation of the neural target map with the gene expression map to obtain the observed r value. (**A.4**) To calculate the significance of the observed r value, we first used BrainSMASH (***Burt et al., 2020***) to simulate 100,000 surrogate maps that preserve the neural target map's spatial autocorrelation. (**A.5**) We then correlated the gene expression map with each of the 100,000 surrogate maps, to generate a distribution of 100,000 simulated r values. (**A.6**) Finally, we estimated the probability of the observed correlation between the neural target map and the gene expression map occurring by chance given the permuted distribution. For further details see (***Figure 2—figure supplement 2***) (**B**) Gene expression pattern for interneuron marker gene somatostatin (SST). Left: positive (yellow) regions show areas where the gene of interest is highly expressed, whereas negative (blue) regions indicate low expression values. Right: bar plot showing the mean gene expression Z-score across association (black) and sensory (white) networks. (**C**) Gene expression pattern for interneuron marker parvalbumin (PVALB). Left: positive (yellow) regions show areas where the gene of interest is highly expressed, whereas negative (blue) regions indicate low expression values. Right: bar plot showing the mean gene expression Z-score across association (black) and sensory (white) networks. (**D**) Unthresholded PC1 Δ GBC Z-score map (Z-scores computed across 718 parcels). Red/orange areas indicate parcels that have a high positive loading score onto PC1, while blue areas indicate parcels that have a high negative loading score onto PC1. (**E**) Bar plot showing the correlation between PC1 Δ GBC and the following gene expression maps: SST (*r*=0.47, p<0.001) (red) and PVALB (*r*=−0.22, p=0.025) (blue). All p-values are FDR corrected. (**F**) Distribution of 100,000 simulated r values for SST (bottom) and PVALB (top). Bold lines indicate the observed r value between PC1 Δ GBC and SST (red) and PVALB (blue). (**G**) Unthresholded mean Δ (ketamine - placebo) GBC Z-score map at the parcel level (No. parcels = 718) (Z-scores computed across 718 parcels). Red/orange areas indicate regions where participants exhibited stronger GBC in the ketamine condition, whereas blue areas indicate regions where participants exhibited reduced GBC in the ketamine condition, compared with the placebo condition. (**H**) Bar plot showing the correlation between mean Δ GBC and the following gene expression maps: SST (*r*=0.15, p=0.363) (red) and PVALB (*r*=−0.06, p=0.627) (blue). All p-values are FDR corrected. (**I**) Distribution of 100,000 simulated r values for SST (bottom) and PVALB (top). Bold lines indicate the observed r value between mean Δ GBC and SST (red) and PVALB (blue).

The online version of this article includes the following figure supplement(s) for figure 3:

**Figure supplement 1.** PC1 Δ GBC and PC3-5 Δ GBC maps tracks SST and PVALB neural gene expression patterns.

**Figure supplement 2.** Gene analysis workflow.

judgement (***Figure 4D***). In contrast, behavioral PC2 indexes variation particularly in regards to: (i) the negative-symptom item poor rapport; (ii) positive-symptom items such as grandiosity, hallucinations, and delusions; (iii) general-symptom items such as uncooperativeness, unusual thought content, impulse control, and preoccupation; and (iv) cognition (***Figure 4D***). See ***Figure 4—figure supplement 1*** for behavioral PC1 and PC2 symptom values. Overall, this indicates that the behavioral effects of acute ketamine administration are also multi-dimensional.

## Lower-dimensional behavioral variation reveals robust neuro-behavioral mapping

In order to relate ketamine's behavioral and neural effects, we mapped the data-reduced ketamine-induced changes in behavior (i.e. behavioral PC1 and PC2) to the neural data (***Ji et al., 2020***). This

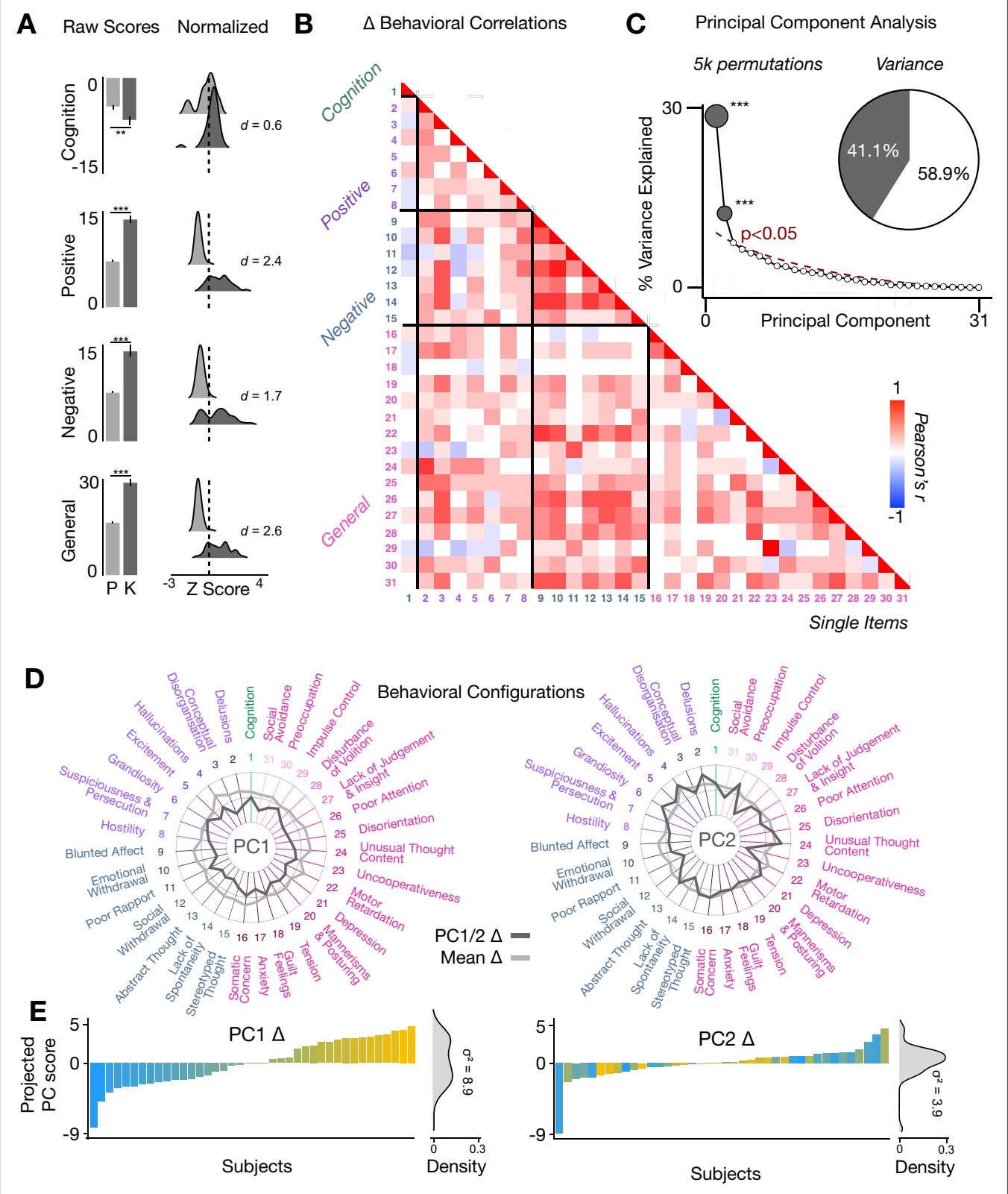

**Figure 4.** Multidimensional behavioral effect of acute ketamine administration. (**A**) Mean raw scores (left panel) and distribution of normalized scores (right panel) for placebo (**P**) and ketamine (**K**) across cognition (spatial working memory) and subjective effects (PANSS positive, PANSS negative, and PANSS general symptoms). Bar plot error bars show standard deviations; distribution plot effect sizes are calculated using Cohen's d. Condition is color coded (light gray = placebo, dark gray = ketamine). (**=p < 0.05, ***=p < 0.001). For further details see **Supplementary file 1**. (**B**) Correlations between

*Figure 4 continued on next page*

*Figure 4 continued*

31 individual item behavioral measures for all participants (N=40). Δ (ketamine - placebo) behavioral measures are used. (**C**) Screeplot showing the % variance explained by each of the principal components (PCs) from a PCA performed using all 31 Δ (ketamine - placebo) behavioral measures across 40 participants. The size of each dark gray point is proportional to the variance explained. The first two PCs (dark gray) survived permutation testing (p<0.05, 5000 permutations). Together they capture 41.1% of all symptom variance (inset), with PC1 explaining 29% of the variance and PC2 explaining 12% of the variance. (**D**) Loading profiles shown in dark gray across the 31 behavioral items for PC1 Δ (left) and PC2 (right) Δ. See *Figure 3—figure supplement 1* for numerical values of the behavioral item scores for each PC. The mean Δ score for each behavioral item is also shown in light gray (scaled to fit the same radar plots). Note that the mean Δ configuration resembles the PC1 Δ loading profile more closely than PC2 Δ (which is to be expected as PC1 explains more variance in the behavioral measures). Inner circle = –0.5, middle circle = 0, outer circle = 0.5. (**E**) Bar plot showing the projected PC score for each individual subject (N=40) for PC1 Δ (left) and PC2 (right) Δ. Bars are color coded according to each subject's PC1 score ranking. The density of the projected PC scores is displayed on the right of each bar plot, demonstrating again that PC1 explains the greatest amount of variance. A paired Pitman-Morgan test revealed that there is a significant difference in variance between the individual subject scores in PC1 Δ and PC2 Δ (t=6.4, df = 38, p<0.001).

The online version of this article includes the following source data and figure supplement(s) for figure 4:

**Figure supplement 1.** Delta Behavioral Principal Component (PCA) Loadings.

**Figure supplement 1—source data 1.**

**Figure supplement 2.** Principal Component Analysis: PANSS, CADSS & Cognition.

**Figure supplement 3.** Principal Component Analysis: CADSS & Cognition.

resulted in two bi-directional neuro-behavioral PCs that are interpretable in relation to behavior: neuro-behavioral PC1 demonstrates that a high positive behavioral PC1 score is associated with increased GBC in association networks (i.e. default mode network, fronto-parietal network) and decreased GBC in sensory networks (i.e. secondary visual network), while a high negative behavioral PC1 score is associated with the opposite pattern (*Figure 5—figure supplement 4B*). In addition, we found both neuro-behavioral PC1 and PC2 differentiate between association and sensory networks (*P*<.001), and neuro-behavioral PC1 tracks SST and PVALB gene expression patterns (*Figure 5A and B*, *Figure 5—figure supplement 4E*, *Figure 5—figure supplement 3A, C, E*). Finally, the neuro-behavioral PCs capture unique patterns of neural variation in comparison to the neural PCs: neuro-behavioral PC1 correlates moderately with PC1 Δ GBC (*r*=−0.61), while neuro-behavioral PC2 correlates weakly with PC1-5 Δ GBC (*Figure 5—figure supplement 2*).

In order to compare the two data-driven lower-dimensional behavioral axes of variation to the established subscales, we then mapped the ketamine-induced changes in PANSS subscales and cognition to the Δ GBC maps (*Figure 5—figure supplement 5*). Overall, we found comparable results when regressing the behavioral PCs and the PANSS subscales onto the neural data. For example, the neuro-behavioral PC1, PANSS Negative, and PANSS General maps are all highly correlated, capture similar amounts of variation, differentiate between association and sensory networks, and are associated with SST and PVALB cortical gene expression patterns (*Figure 5*, *Figure 5—figure supplement 7*, *Figure 5—figure supplement 3*).

Finally, as the PANSS literature has indicated that a five-factor model may be more stable than the original three-factor model, we also repeated this analysis using the following five PANSS factors: Positive, Negative, Disorganization, Excitement, and Emotional Distress (see *Figure 5—figure supplement 8*; *van der Gaag et al., 2006*). Overall, the three-factor and five-factor PANSS yielded comparable results: the five-factor PANSS Negative and five-factor PANSS Disorganization neural maps are also highly correlated with neuro-behavioral PC1 (*Figure 5—figure supplement 8*). However, neuro-behavioral PC2 showed a much stronger positive correlation with the five-factor PANSS Positive map (*r*=0.73) than it did with the three-factor PANSS Positive map (*r*=0.42; *Figure 5—figure supplement 6B*, *Figure 5—figure supplement 8C*).

## Neuro-behavioral mapping captures individual variation in how SST and PVALB interneurons may be differentially impacted by ketamine

To explore whether the lower-dimensional behavioral axes of variation can lead to inferences about ketamine's neural effects, we: (i) compared inter-individual variability in behavioral PC1 and neuro-behavioral PC1; (ii) explored whether neuro-behavioral PC1 captures inter-individual variability in how ketamine's molecular mechanisms may relate to its neural and behavioral effects.

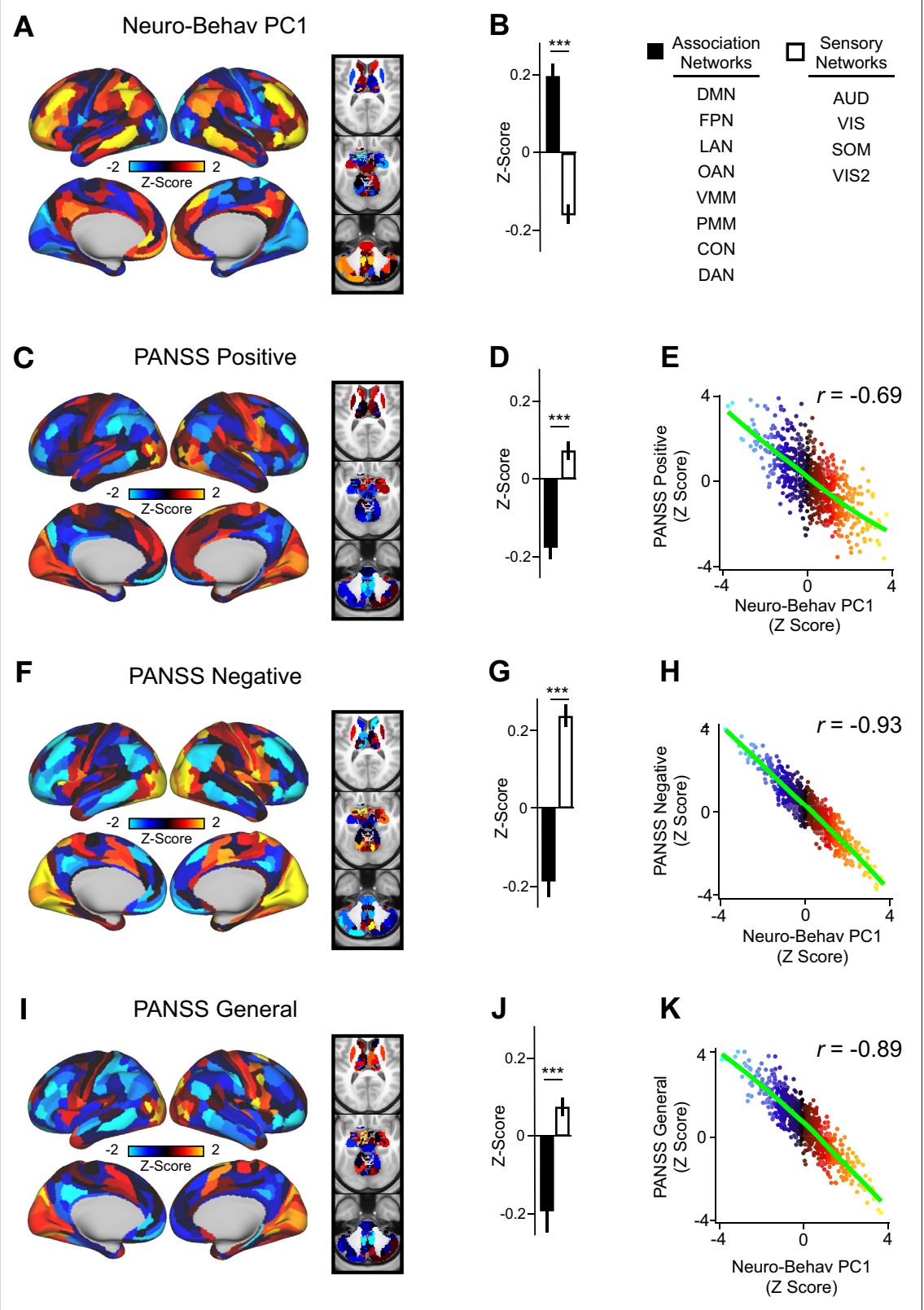

**Figure 5.** Lower-dimensional behavioral variation reveals robust neuro-behavioral mapping. (**A**) Neuro-behavioral PC1 map showing the relationship between the behavioral PC1 score for each participant regressed onto the Δ GBC map for each participant (N=40). Values shown in each brain parcel are the Z-scored regression coefficient (behavioral PC1 score, Δ GBC) across all 40 subjects. Red/orange areas indicate parcels in which there is a positive relationship between GBC and the behavioral PC1 score, while blue areas indicate parcels in which there is a negative relationship between

*Figure 5 continued on next page*

Figure 5 continued

GBC and the behavioral PC1 score. (**B**) Bar plot showing the mean correlation (Δ GBC, behavioral PC1 score) for association (black) and sensory (white) networks. (**C**) PANSS Positive map showing the relationship between the PANSS Positive score for each participant regressed onto the Δ GBC map for each participant (N=40). Values shown in each brain parcel are the Z-scored regression coefficient (PANSS Positive score, Δ GBC) across all 40 subjects. (**D**) Bar plot showing the mean correlation (Δ GBC, PANSS Positive score) for association (black) and sensory (white) networks. (**E**) Scatter plot showing the relationship across parcels between neuro-behavioral PC1 and PANSS Positive maps (*r*=0.69, p<0.001, n=718). (**F**) PANSS Negative map showing the relationship between the PANSS Negative score for each participant regressed onto the Δ GBC map for each participant (N=40). Values shown in each brain parcel are the Z-scored regression coefficient (PANSS Negative score, Δ GBC) across all 40 subjects. (**G**) Bar plot showing the mean correlation (Δ GBC, PANSS Negative score) for association (black) and sensory (white) networks. (**H**) Scatter plot showing the relationship across parcels between neuro-behavioral PC1 and PANSS Negative maps (*r*=0.93, p<0.001, n=718). (**I**) PANSS General map showing the relationship between the PANSS General score for each participant regressed onto the Δ GBC map for each participant (N=40). Values shown in each brain parcel are the Z-scored regression coefficient (PANSS General score, Δ GBC) across all 40 subjects. (**J**) Bar plot showing the mean correlation (Δ GBC, PANSS General score) for association (black) and sensory (white) networks. (**K**) Scatter plot showing the relationship across parcels between neuro-behavioral PC1 and PANSS General maps (*r*=0.89, p<0.001, n=718). (DMN = default mode, FPN = frontoparietal, LAN = language, OAN = orbito-affective, VMM = ventro multimodal, PMM = posterior multimodal, CON = cingulo-opercular, DAN = dorsal attention, AUD = auditory, VIS = primary visual, SOM = somatomotor, VIS2=secondary visual).

The online version of this article includes the following figure supplement(s) for figure 5:

**Figure supplement 1.** Comparison of three principal component analyses: 1. PANSS & Cognition; 2. PANSS, CADSS, & Cognition;and 3. CADSS, & Cognition.

**Figure supplement 2.** Correlations between PC1-5 Δ GBC, neuro-behavioral PC1-2, and mean Δ GBC maps.

**Figure supplement 3.** Neuro-behavioral PC1, PANSS negative, and PANSS general track SST and PVALB gene expression to-pographies.

**Figure supplement 4.** Network breakdown of neuro-behavioral PC1 and neuro-behavioral PC2.

**Figure supplement 5.** Network breakdown of existing behavioral measures: 3-factor PANSS subscales and cognition Δ GBC maps.

**Figure supplement 6.** Scatter plots showing the relationship between neuro-behavioral PC1-2 and the 3-factor PANSS subscales.

**Figure supplement 7.** Individual variation captured by neuro-behavioral PCs, 3-factor PANSS subscales, and cognition Δ GBCmaps.

**Figure supplement 8.** Scatter plots showing the relationship between neuro-behavioral PC1-2 and the 5-factor PANSS subscales.

To compare inter-individual variability in behavioral PC1 and neuro-behavioral PC1, we plotted the rank-ordering of: (i) each individual subject's behavioral PC1 score; and (ii) each individual subject's neuro-behavioral PC1 score (*Figure 6A–B*). The behavioral PC1 scores were highly correlated with the neuro-behavioral PC1 scores (*r*=0.7, p<0.001), indicating that whether individual variation is captured using behavior or behavior in relation to the neural effect, subjects largely preserve their order (*Figure 6C*). We then assessed whether, like neural PC1 and 3–5, neuro-behavioral PC1-2 are related to SST and PVALB gene expression maps using the method outlined in *Figure 3A*. We found that neuro-behavioral PC1 but not neuro-behavioral PC2 showed a distinct relationship with SST vs. PVALB gene expression maps: neuro-behavioral PC1 showed a significant positive correlation with SST (*r*=0.29, p=0.009), and non-significant negative correlation with PVALB (*r*=−0.14, p=0.089; *Figure 6E–F*).

Given neuro-behavioral PC1 represents a bi-directional axes of variation, we wanted to explore whether it captures individual variation in how ketamine-induced neuro-behavioral changes differ-entially relate to SST and PVALB cortical gene expression maps. To explore this, we first calculated how each individual participant's Δ GBC map relates to SST and PVALB cortical gene expression profiles, again using the method outlined in *Figure 3A*. We observed that 35/40 participants exhibit a distinct association with SST and PVALB cortical gene expression maps: 23 participants show a positive correlation with SST cortical gene expression maps and a positive negative correlation with PVALB cortical gene expression maps, while 12 participants show the reverse pattern (*Figure 6— figure supplements 1–4*).

We then plotted each participant's observed SST and PVALB r values (generated by correlating their Δ GBC map with the SST and PVALB cortical gene expression maps), ordering the participants according to their neuro-behavioral PC1 score (assessed by calculating the similarity of an indi-viduals' Δ GBC map to the neuro-behavioral PC1 map; *Figure 6G*). The revealed that a negative neuro-behavioral PC1 score broadly corresponds to a negative association with the SST cortical gene expression map and a positive association with the PVALB cortical gene expression map (*Figure 6G*). Conversely, a positive neuro-behavioral PC1 score predominantly signifies the opposite pattern (*Figure 6G*). When we formally tested this association by correlating each participant's difference in

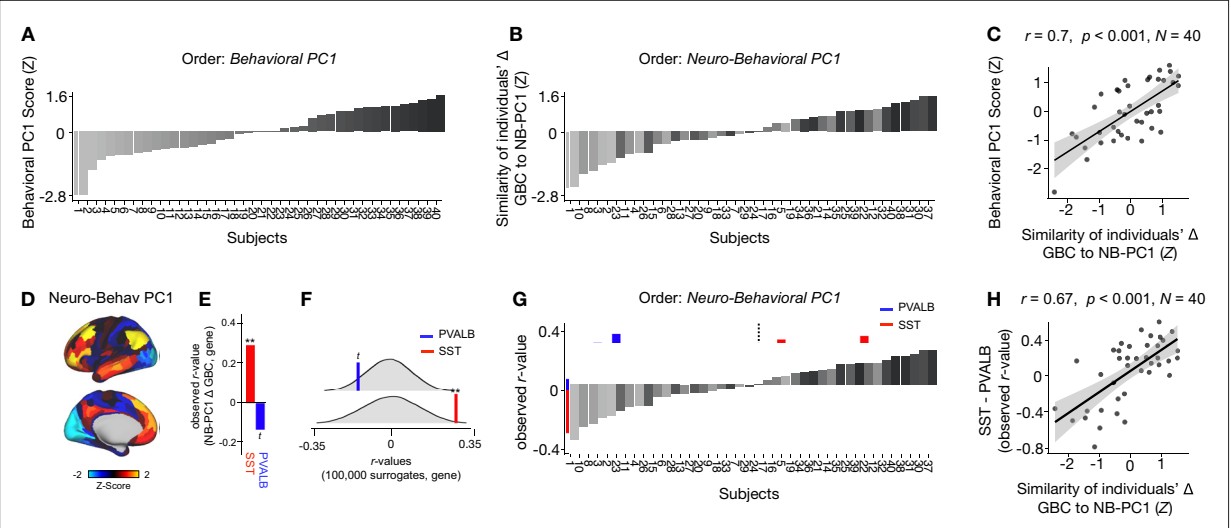

**Figure 6.** Individual variation in ketamine-induced neuro-behavioral changes. (**A**) Bar plot showing the behavioral PC1 score (Z) for each individual participant (N=40). Bars are numbered and color-coded according to each participant's behavioral PC1 score (light gray = highly negative score, dark gray = highly positive score). (**B**) Bar plot showing the neuro-behavioral PC1 score (Z) for each individual participant (N=40). The neuro-behavioral PC1 score is calculated by correlating each participant's Δ GBC map with the neuro-behavioral PC1 map, and then Z-scoring the r-values. Bars are ordered according to each participant's neuro-behavioral PC1 score, but labelled and color-coded according to each participant's behavioral PC1 score (light gray = h ighly negative score, dark gray = highly positive score). (**C**) Correlation between each participant's behavioral PC1 score and their neuro-behavioral PC1 score (r=0.7, p<0.001). (**D**) Neuro-behavioral PC1 map showing the relationship between the behavioral PC1 score for each participant regressed onto the Δ GBC map for each participant (N=40). (**E**) Bar plot showing the correlation between the neuro-behavioral PC1 map and the following gene expression maps: SST (r=0.29, p=0.009) (red) and PVALB (r=−0.14, p=0.089) (blue). All p-values are FDR corrected. (**F**) Distribution of 100,000 simulated r values for neuro-behavioral PC1 and: SST (bottom), PVALB (top). Bold lines indicate the observed r value between neuro-behavioral PC1 and: SST (red), PVALB (blue). (**G**) Bar plot showing the relationship between SST (red) and PVALB (blue) gene expression maps and each participants Δ GBC maps (i.e. the observed r-value). Participants are ordered according to their neuro-behavioral PC1 score (i.e. the similarity between their Δ GBC map and the neuro-behavioral PC1 map). Subjects to the left of the dashed line load negatively onto neuro-behavioral PC1, while subjects to the right of the dashed line load positively onto neuro-behavioral PC1. (**H**) Correlation between the difference (SST - PVALB) in observed r values for each participant, and their neuro-behavioral PC1 score (r=0.67, p<0.001).

The online version of this article includes the following figure supplement(s) for figure 6:

**Figure supplement 1.** Relationship between Δ GBC maps and SST/PVALB gene expression maps for subjects 1-10.

**Figure supplement 2.** Relationship between Δ GBC maps and SST/PVALB gene expression maps for subjects 11-20.

**Figure supplement 3.** Relationship between Δ GBC maps and SST/PVALB gene expression maps for subjects 21-30.

**Figure supplement 4.** Relationship between Δ GBC maps and SST/PVALB gene expression maps for subjects 31-40.

observed SST and PVALB r values with their neuro-behavioral PC1 score, we found a significant positive relationship (r=0.67, p<0.001; *Figure 6H*).

## Discussion

Ketamine has emerged as one of the most promising treatments for depression (*Krystal et al., 2019*; *Murrough et al., 2013*). However, inter-individual variability in behavioral and neural responses to acute ketamine administration has been largely ignored. This study addresses this knowledge gap by showing that: (i) both the neural and behavioral effects of acute ketamine are multi-dimensional, reflecting robust inter-individual variability; (ii) the use of a multi-dimensional vs. uni-dimensional neural approach is more closely associated with ketamine's molecular mechanisms; and (iii) ketamine's data-driven principal behavioral gradient provides an anchor point for ketamine's neural effects and molecular mechanisms at the single subject level.

## Ketamine's neural effects are multi-dimensional, reflecting inter-individual variability

We show that the neural geometry of ketamine is multi-dimensional, as ketamine results in five significant neural axes of variation, each of which is bi-directional (*Figure 2A*, *Figure 2—figure supplement 3*). For example, following ketamine high positive PC1 Δ GBC loaders exhibit increased GBC in association networks such as the default mode network and decreased GBC in sensory cortices such as the secondary visual network (i.e. subject 28 in *Figure 6—figure supplement 3H*), while high negative PC1 Δ GBC loaders show the reverse pattern (i.e. subject 32 in *Figure 6—figure supplement 4B*). The bi-directional nature of the PCs helps explain the contradictory results in the ketamine resting-state literature as the direction of effects is often inconsistent (*Höflich et al., 2015*; *Hack et al., 2021*; *Grimm et al., 2015*; *Khalili-Mahani et al., 2015*; *Kraguljac et al., 2017*).

Interpreting this finding is of particular interest given that ketamine is a molecule that acts at the same target in every human brain, antagonising the NMDA receptor, which should intuitively result in a single uni-directional principal component (PC). A number of possible interpretations exist for why ketamine results in multiple bi-directional neural PCs. For example, one PC may reflect ketamine's effects in excitatory-excitatory synapses, while the another may reflect ketamine's effects in excitatory-inhibitory synapses. Another possibility is that, given we know ketamine's affinity for certain cells is based on their NMDA receptor subunit configuration, one PC may reflect ketamine's 'high affinity' actions on GluN2C and GluN2D containing receptors, while another PC may reflect ketamine's 'lower affinity' actions on GluN2A and GluN2B containing receptors (*Kotermanski and Johnson, 2009*; *Kuner and Schoepfer, 1996*; *Khlestova et al., 2016*). A third possibility is that the multiple bi-directional PCs may be driven by ketamine's numerous microcircuit level targets, as ketamine effects: (i) SST interneurons, which in turn disinhibit glutamatergic pyramidal distal dendrites and PVALB interneurons; (ii) PVALB interneurons, which disinhibit the soma of glutamatergic pyramidal neurons; (iii) VIP neurons, which may reduce distal dendrite inhibition but disinhibit SST activity; and (iv) chandelier cells, which permit the back-propagation of action potentials from the axon to the soma *Anderson et al., 2020*. Meanwhile, the bi-directional nature of the principal components may relate to individual differences in a number of factors including use-dependent excitation of NMDA networks, genetic variation in NMDA receptor subunits, and sleep profiles *MacDonald et al., 1987*; *Xu and Lipsky, 2015*; *De Vry and Jentzsch, 2003*; *Duncan et al., 2013*.

Ketamine's numerous microcircuit level targets may also explain why ketamine shows increased dimensionality compared to psilocybin and LSD: psilocybin and LSD's behavioral effects are predominantly linked to 5HT2A agonism, and 5HT2A receptors are located on fewer cellular elements (*Figure 2 Winter et al., 2007*; *Benneyworth et al., 2005*; *Preller et al., 2017*; *Vollenweider et al., 1998*). In addition, ketamine's multi-dimensional neural effects may be due to complementary inhibitory and excitatory dimensions which are not present to the same extent with psilocybin and LSD: both psilocybin and LSD are excitatory while ketamine results in increased excitation indirectly as a consequence of the inhibition of interneurons (*Nichols and Nichols, 2021*; *Gerhard et al., 2020*).

## Ketamine's data-driven principal neural gradient captures is associated with its hypothesized molecular mechanisms while the mean effect is not

We found ketamine's data-driven principal neural gradient is associated with SST and PVALB cortical gene expression maps, as are PC3-5 Δ GBC (*Figure 3*, *Figure 3—figure supplement 1*). In contrast, no significant relationship was found between ketamine's mean effect and SST/PVALB cortical gene expression maps (*Figure 3H*). Given the relationship between ketamine's hypothesized SST and PVALB molecular mechanisms of action and its neural effects is only evident when using a multi-dimensional approach, this is an important consideration for future studies.

## Ketamine results in two data-driven principal behavioral gradients that capture novel neural variation

We found that ketamine's behavioral effects are multi-dimensional, as ketamine resulted in two significant bi-directional behavioral axes of variation (*Figure 4*). This demonstrates that there is also inter-individual variation in the behavioral response to ketamine that is not fully captured by the mean. For

example, while behavioral PC1 is closely related to the mean effect, behavioral PC2 tracks changes in measures such as 'uncooperativeness' and 'impulse control' that are not captured by the mean (*Figure 4D*). This finding is also of interest in relation to the wider literature, as most data-reduction PANSS studies have identified a five-factor rather than a two-factor solution (*van der Gaag et al., 2006*). However, previous PANSS data-reduction studies focused on psychosis populations rather than acute ketamine administration.

The two data-driven principal behavioral gradients capture novel neural variation when regressed onto the Δ GBC data, resulting in two neuro-behavioral PCs. The novel neural variation captured by the neuro-behavioral PCs is evidenced by the fact that neuro-behavioral PC2 and PC2 Δ GBC were only weakly correlated (*Figure 5—figure supplement 2*). Additionally, in contrast to PC1-5 Δ GBC, the neuro-behavioral PCs are interpretable in relation to behavior. Such an understanding is crucial for informing predictions about how a specific individual will respond to ketamine in a clinical setting. Importantly, this prediction may ultimately be done using only their symptom profile, without the need for more costly neural scans.

Finally, when we directly compared ketamine's data-driven principal behavioral gradient to the existing PANSS subscales, we found that regressing the two behavioral PCs, the three-factor PANSS subscales, and the five-factor PANSS subscales onto the Δ GBC data produced comparable results (*Figure 5—figure supplement 6*, *Figure 5—figure supplement 8*, *Figure 5—figure supplement 3*). This highlights the validity of the existing PANSS subscales.

## Ketamine's data-driven principal behavioral gradient provides an anchor point for neural effects and molecular mechanisms at the single subject level

The ultimate goal for developing a framework for connecting across multiple levels of analysis is to be able take an individual's behavioral response to a molecular perturbation, and precisely predict their imaging effect, or vice versa. When we tested our neuro-behavioral model by comparing the order in which individual participants load onto behavioral PC1 and neuro-behavioral PC1, we found the order was largely preserved (*Figure 6A–C*). Thus, this framework enables us to take an individual's behavioral response to ketamine, and then meaningfully infer where that individual falls in relation to their ketamine-induced neural effects. Given this framework operates at the whole-brain level, this indicates that the neural effect of ketamine is not ROI-based but a distributed system-wide effect.

Furthermore, we demonstrated that neuro-behavioral PC1 captured individual variation in how SST and PVALB interneurons may be differentially impacted by ketamine: we found an association between participants' neuro-behavioral PC1 scores and whether their Δ GBC maps positively correlated with SST cortical gene expression maps and negatively correlated with PVALB cortical gene expression maps (seen in 58% of participants), or whether they showed the reverse pattern (seen in 12% of participants) (*Figure 6H*). A potential interpretation is that the expression of SST and PVALB cortical genes varies among individuals due to a range of factors, including stress, sleep patterns, medications, and psychiatric diagnoses (*Lin and Sibille, 2015*; *Perez et al., 2019*). Overall, this is an important intuition, as the identification of the targets responsible for the different behavioral effects of ketamine seen in specific individuals is critical for the development of novel pharmaco-therapies that may lack the side effects of ketamine or reduce the abuse potential.

## Limitations and future directions

As with most psychoactive pharmacological neuroimaging studies, despite the single-blind design participants were able to correctly identify whether they received ketamine or saline 100% of the time. Future studies may mitigate this by using an active placebo control. In addition, as the ketamine and saline scans took place on the same day, ketamine's residual effects ruled out the counterbalancing of the order. Furthermore, neural measures were only collected during ketamine's peak effects. Given ketamine's antidepressant effects are shown to peak 24–72 hr post-infusion, future studies may wish to collect additional follow-up scans (*Krystal et al., 2019*; *Berman et al., 2000*). While behavioral measures were collected during both peak effects and offset, the use of repetitive questionnaires and tasks introduces potential psychometric confounds such as familiarity and practice effects. In addition, we did not regress out physiological measures. There is a large body of evidence to suggest that heart rate and blood pressure impact resting-state functional connectivity, and sub-anesthetic doses

of ketamine and moderate doses of psilocybin and LSD have all been shown to increase heart rate and blood pressure (*Chang et al., 2013*; *Nikolaou et al., 2016*; *de la Cruz et al., 2019*; *Riva-Posse et al., 2018*; *Bennett et al., 2023*; *Isbell, 1959*). While to date no studies have directly compared the physiological effects of all three drugs, there is evidence that LSD and psilocybin have distinct effects on blood pressure and heart rate (*Holze et al., 2022*).

The present study used GBC: a dimension-reduced summary measure of neural resting-state fMRI. While this remains a principled method of reducing the feature space, it is possible that some of the behaviorally relevant neural information may be lost by first summarizing neural features in this manner. Thus, future studies may wish to explore the multi-dimensional effects of ketamine using the full neural functional connectivity matrix. In this analysis, we largely focused on PC1-2 Δ GBC when exploring ketamine's neural effects as they explained the bulk of the neural variance. To unpack the variation in PC3-5 Δ GBC a larger sample size is needed, especially given each PC represents a bidirectional axis of variation. A larger sample size would also allow us to investigate alternative models for ketamine's molecular mechanisms in addition to the indirect hypothesis, such as how ketamine's effects may differ based on NMAR subtypes or excitatory vs. inhibitory synapses. It would also enable us to explore whether any demographic characterises such as sex relate to specific PCs. Furthermore, given the differences in effective dimensionality we see across different drugs, an important future direction will be an expansion of the primary analysis to LSD and psilocybin in order to more thoroughly compare their effects. Finally, it will be important for future studies to assess the importance of intra-individual variation in addition to inter-individual variation.

## Conclusions

A key goal in psychiatric research is to predict treatment response. One of the main barriers to the development of predictive biomarkers for treatment response to ketamine is our limited understanding of inter-individual variability in ketamine response. We address this knowledge gap by: (i) showing there is robust inter-individual variability in both the behavioral and neural response to ketamine that requires a multi-dimensional analytic approach; and (ii) providing a multi-dimensional framework with which to connect across ketamine's behavioral, neural and molecular effects that is resolvable at the single subject level. This multi-dimensional framework has the potential to generate predictions about how an individual will respond to ketamine, and as such the crucial next step will be to test this framework in patients using actual treatment response data.

## Acknowledgements

We would like to thank Dr. Robert Malison for his contribution to the conceptualization of the study and analyses. We would also like to thank the Yale Magnetic Resonance Research Center (MRRC) and the Yale Center for Clinical Investigation (YCCI). This study was supported by NIH grants DP5OD012109 (AA), U01MH121766 (AA), R01MH112746 (JDM), R01MH108590 (AA), R01MH112189 (AA), NIAAA grant 2P50AA012870-11 (AA); NSF NeuroNex grant 2015276 (JDM); Brain and Behavior Research Foundation Young Investigator Award (AA); SFARI Pilot Award (JDM, AA); Heffter Research Institute (Grant No. 1–190420); Swiss Neuromatrix Foundation (Grant No. 2016–0111 m Grant No. 2015 – 010); Swiss National Science Foundation under the framework of Neuron Cofund (Grant No. 01EW1908), Usona Institute (2015 – 2056).

## Additional information

### Competing interests

Jie Lisa Ji: J.L.J. is an employee of Manifest Technologies and has previously worked for Neumora (formerly BlackThorn Therapeutics) and is a co-inventor on the following patent: Anticevic A, Murray JD, Ji JL: Systems and Methods for Neuro-Behavioral Relationships in Dimensional Geometric Embedding (N-BRIDGE), PCT International Application No. PCT/US2119/022110, filed March 13, 2019. Joshua B Burt: Currently an employee of Neumora Therapeutics and consulted for BlackThorn Therapeutics in 2019. Zailyn Tamayo, Clara Fonteneau: Consults for Manifest Technologies and previously consulted for Neumora (formerly BlackThorn Therapeutics). Grega Repovs: Consults for and holds

equity with RBNC (formerly BlackThorn Therapeutics). Sarah K Fineberg: Sarah K. Fineberg: discloses work with the pharmaceutical company Boehringer Ingelheim as site PI for a multinational clinical trial and for consulting on advisory boards (< $10,000 in 2022). John H Krystal: Was a consultant for Aptinyx, Atai Life Sciences, AstraZeneca Pharmaceuticals, Biogen Idec, Biomedisyn Corporation, Bionomics, Limited, Boehringer Ingelheim International, Cadent Therapeutics, Clexio Bioscience, COMPASS Pathways, Limited, Concert Pharmaceuticals, Epiodyne, EpiVario, Greenwich Biosciences, Heptares Therapeutics, Limited (UK), Janssen Research \& Development, Jazz Pharmaceuticals, Otsuka America Pharmaceutical, Perception Neuroscience Holdings, Spring Care, Sunovion Pharmaceuticals, Takeda Industries, Taisho Pharmaceutical Co. He served on advisory boards for Biohaven Pharmaceuticals, BioXcel Therapeutics, BlackThorn Therapeutics, Cadent Therapeutics, Cerevel Therapeutics, EpiVario, Eisai, Lohocla Research Corporation, Novartis Pharmaceuticals Corporation, PsychoGenics, Tempero Bio, and Terran Biosciences. He owns stock or stock options in Biohaven Pharmaceuticals, Sage Pharmaceuticals, Spring Care, Biohaven Pharmaceuticals Medical Sciences, BlackThorn Therapeutics, EpiVario, Terran Biosciences, and Tempero Bio, and he is on the editorial board of Biological Psychiatry. John D Murray: Consults for and holds equity with RBNC (formerly BlackThorn Therapeutics) and is a coinventor on the following patents: Anticevic A, Murray JD, Ji JL: Systems and Methods for Neuro-Behavioral Relationships in Dimensional Geometric Embedding (N-BRIDGE), PCT International Application No. PCT/US2119/022110, filed March 13, 2019 and Murray JD, Anticevic A, Martin, WJ: Methods and tools for detecting, diagnosing, predicting, prognosticating, or treating a neurobehavioral phenotype in a subject, U.S. Application No. 16/149,903 filed on October 2, 2018, U.S. Application or PCT International Application No. 18/054,009 filed on October 2, 2018. Katrin H Preller: Currently an employee of by Boehringer Ingelheim GmBH \& CO KG. Alan Anticevic: Consults for and holds equity with RBNC (formerly BlackThorn Therapeutics) and is a coinventor on the following patents: Anticevic A, Murray JD, Ji JL: Systems and Methods for Neuro-Behavioral Relationships in Dimensional Geometric Embedding (N-BRIDGE), PCT International Application No. PCT/US2119/022110, filed March 13, 2019 and Murray JD, Anticevic A, Martin, WJ: Methods and tools for detecting, diagnosing, predicting, prognosticating, or treating a neurobehavioral phenotype in a subject, U.S. Application No. 16/149,903 filed on October 2, 2018, U.S. Application for PCT International Application No. 18/054,009 filed on October 2, 2018. The other authors declare that no competing interests exist.

## Funding

| Funder | Grant reference number | Author |
| --- | --- | --- |
| National Institutes of Health | DP5OD012109-01 | Alan Anticevic |
| National Institutes of Health | R01MH112746 | John D Murray |
| National Institutes of Health | 5R01MH112189 | Alan Anticevic |
| National Institutes of Health | 5R01MH108590 | Alan Anticevic |
| National Institute on Alcohol Abuse and Alcoholism | 2P50AA012870-11 | Alan Anticevic |
| National Science Foundation | 2015276 | John D Murray |
| Brain and Behavior Research Foundation | Young Investigator Award | Alan Anticevic |
| Simons Foundation Autism Research Initiative | Pilot Award | John D Murray Alan Anticevic |
| Heffter Research Institute | 1-190420 | Franz X Vollenweider Katrin H Preller |
| Swiss Neuromatrix Foundation | 2016-0111 | Franz X Vollenweider Katrin H Preller |

| Funder | Grant reference number | Author |
| --- | --- | --- |
| Swiss National Science Foundation under the framework of Neuron Cofund | 01EW1908 | Katrin H Preller |
| Usona Institute | 2015 - 2056 | Franz X Vollenweider |
| National Institutes of Health | 1U01MH121766 | Alan Anticevic |

The funders had no role in study design, data collection and interpretation, or the decision to submit the work for publication.

## Author contributions

Flora Moujaes, Conceptualization, Data curation, Formal analysis, Validation, Visualization, Methodology, Writing – original draft, Writing – review and editing; Jie Lisa Ji, Conceptualization, Software, Formal analysis, Supervision, Visualization, Methodology, Writing – original draft, Writing – review and editing; Masih Rahmati, Software, Formal analysis, Methodology, Writing – review and editing; Joshua B Burt, Data curation, Software, Formal analysis, Investigation, Methodology, Project administration; Charles Schleifer, Brendan D Adkinson, Resources, Data curation, Investigation, Project administration, Writing – review and editing; Aleksandar Savic, Resources, Data curation, Software, Investigation, Methodology, Project administration; Nicole Santamauro, Project administration, Writing – review and editing; Zailyn Tamayo, Resources, Data curation, Software, Methodology, Project administration, Writing – review and editing; Caroline Diehl, Resources, Project administration; Antonija Kolobaric, Resources, Software, Investigation, Methodology, Project administration, Writing – review and editing; Morgan Flynn, Resources, Data curation; Nathalie Rieser, Visualization, Writing – review and editing; Clara Fonteneau, Terry Camarro, Supervision, Funding acquisition, Investigation, Writing – review and editing; Junqian Xu, Resources, Software, Methodology; Youngsun Cho, Conceptualization, Resources, Software, Formal analysis, Supervision, Funding acquisition, Validation, Investigation, Visualization, Methodology, Writing – original draft, Project administration, Writing – review and editing; Grega Repovs, Conceptualization, Resources, Data curation, Software, Formal analysis, Supervision, Funding acquisition, Validation, Investigation, Visualization, Methodology, Writing – original draft, Project administration, Writing – review and editing; Sarah K Fineberg, Alan Anticevic, Conceptualization, Resources, Data curation, Software, Formal analysis, Supervision, Funding acquisition, Investigation, Visualization, Methodology, Project administration, Writing – review and editing; Peter T Morgan, Funding acquisition, Investigation, Project administration, Writing – review and editing; Erich Seifritz, Funding acquisition, Writing – review and editing; Franz X Vollenweider, Supervision, Writing – review and editing; John H Krystal, Resources, Software, Investigation, Visualization, Methodology, Writing – review and editing; John D Murray, Conceptualization, Resources, Software, Formal analysis, Supervision, Funding acquisition, Validation, Investigation, Visualization, Methodology, Writing – review and editing; Katrin H Preller, Conceptualization, Supervision, Funding acquisition, Validation, Visualization, Methodology, Writing – review and editing

## Author ORCIDs

Flora Moujaes ⓘ https://orcid.org/0000-0002-0843-0393
Jie Lisa Ji ⓘ https://orcid.org/0000-0002-6280-9070
Charles Schleifer ⓘ http://orcid.org/0000-0001-8411-0968
Brendan D Adkinson ⓘ http://orcid.org/0000-0003-3196-8674
Franz X Vollenweider ⓘ https://orcid.org/0000-0001-9053-6164
John D Murray ⓘ https://orcid.org/0000-0003-4115-8181
Katrin H Preller ⓘ https://orcid.org/0000-0003-0413-7672
Alan Anticevic ⓘ https://orcid.org/0000-0002-4324-0536

## Ethics

ClinicalTrials.gov Identifier: NCT03842800.
The protocol was approved by the Yale Human Investigations Committee (ClinicalTrials.gov Identifier: NCT03842800). All subjects provided written informed consent.

Decision letter and Author response
Decision letter https://doi.org/10.7554/eLife.84173.sa1
Author response https://doi.org/10.7554/eLife.84173.sa2

## Additional files

### Supplementary files
• Supplementary file 1. Retrospectively assessed (180 mins after drug administration) ketamine-induced subjective effects. Effects were assessed using the Clinician Administered Dissociative States Scale (CADSS), the Positive and Negative Syndrome Scale for positive symptoms, negative symptoms and general psychopathology, and the Beck's Depression Inventory (BDI). N=40. ** indicates $P<.001$

• Supplementary file 2. Demographic Information.

• Supplementary file 3. Difference in PANSS Total for each individual participant following ketamine vs. placebo administration. For each participant we conducted a paired t-test comparing their PANSS Total score following either ketamine for placbeo administration. * significant at $P<.05$ uncorrected.

• Supplementary file 4. Number of outliers for each PANSS Item and Cognition Δ (ketamine - placebo) scores. * these items have an interquartile range of 0 so any scores above or below 0 are defined as outliers. Outlier is defined as <Q1-1.5*IQR/>Q3+1.5*IQR where IQR = interquartile range, Q1=first quartile, and Q3=third quartile.

• MDAR checklist

### Data availability
Data presented in this paper are available at https://github.com/AnticevicLab/ketamine_manuscript, (copy archived at *AnticevicLab, 2024*).

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
