## [Editor Report]

This paper presents a valuable investigation of the acute effects of ketamine in healthy volunteers using rating scale-based behavioral and resting-state fMRI-based neural measures. It provides solid evidence of inter-individual variability in responses to ketamine along with associations to relevant gene expression markers (somatostatin, parvalbumin) in the brain.

---

## [Decision Letter]

**Decision letter after peer review:**

Thank you for submitting your article "Ketamine induces multiple individually distinct whole-brain functional connectivity signatures" for consideration by *eLife*. Your article has been reviewed by 2 peer reviewers, and the evaluation has been overseen by a Reviewing Editor and Jonathan Roiser as the Senior Editor. The following individual involved in the review of your submission has agreed to reveal their identity: Christoph Kraus (Reviewer #2).

Essential Revisions (for the authors):

– The δ GBC PC maps look identical to the PC maps obtained in each condition (ketamine, placebo) separately (Figure S1), and for LSD and PSB (Figure S6). This essentially implies that none of the findings are specific to the difference between ketamine and placebo, but instead reflect a general feature of the methodology used here. Please report correlations between these maps or provide some other demonstration that the reported findings are indeed specific to the ketamine-placebo contrast.

– LSD and PSB are only used for evaluating dimensionality. A more convincing analysis would compare the primary effects across all drugs to identify the specific effects of ketamine.

– There is no assessment of the specificity of the gene expression correlations. Many different genes show similar spatial patterns. In fact, the PC maps are patterns along the classical sensory fugal hierarchy, which some of these authors have shown to be the dominant PC of gene expression (Burt, Nat Neuro), meaning that many different genes will have similar expression profiles. Please provide stronger evidence supporting the specificity of the effects on the selected genes with respect to appropriate null models (e.g., Wei et al., Hum Brain Mapp, 2022). This is critical to support the main claims associated with this analysis. This analysis should be accompanied by a more detailed explanation of why the specific genes were chosen, as many other choices are possible.

– The discrepancy between mean and PC approaches is expected if a significant variance is explained by >1 PC, so this is not particularly surprising. The rationale for the emphasis on this result is unclear

– The analysis of high and low loaders, which occupies significant volumes of text, is highly qualitative. Please provide full distributions of relevant metrics across all participants and statistical analysis of the entire sample.

– Many different analyses are presented and the figure captions are dense. The readability of the paper could be improved if the results were more streamlined, particularly since it is based on a small number of participants. Some suggestions:

– Remove comparisons with mean GBC and focus on the PCs;

– Focus on the neuro-behavioral results and genetic mapping.

– It would be helpful to see the distribution of the ketamine response on the behavioral variables, frequently the range of this response is small in healthy volunteers and makes analyses difficult. Are there participants who don't have a significant response or outliers?

– Were the physiological measures recorded during or after the infusions? Typically heart rate and blood pressure increase during acute ketamine administration and the same is likely true for acute LSD or psilocybin. Please provide an indication of how these changes affect the GBC

– Please clarify how double-blinding was implemented precisely.

– There might be other sources for inter-individual variation of ketamine's response such as use-dependent excitation of NMDA networks, sleep profiles or genetic variation of NMDA receptor subunits. The authors may want to incorporate these into their explanation of inter-individual variability.

– Please explain in more detail why the PANSS or δ-PANSS (Post-Pre) was used for the PCA. Why not CADSS, which seemed to have stronger effects than the PANSS (Table S1)?

– Please discuss the downsides of using repetitive rating scales in such an endeavor. To this end, I would be interested in the author's view on what explains more variance in neuropsychological effects of repetitive ketamine/lsd/psilocybin experiments: visual analog scales, questionnaires, or tasks.

– How were sex and age effects addressed in the main results?

– Effective dimensionality was calculated using the participation ratio and dataset re-sampling was used to control for sample size in this calculation, but dimensionality is also affected by motion within the sample among other noise sources, which are not well discussed. In particular, each drug may affect physiological noise in different ways and this may in turn affect their dimensionality measurement. A more detailed comparison across conditions of the effects of motion and the global signal is required. For instance, the authors could compare FD statistics and properties of the global signal (e.g., variance) and physiological recordings.

– On a related point, the description of the FD cutoff for scrubbing frames requires more detail. Is the median taken across all scans and all participants? Or all scans within each participant separately? What was the final cutoff determined by this procedure?

---

## [Author Response]

Essential Revisions (for the authors):– The δ GBC PC maps look identical to the PC maps obtained in each condition (ketamine, placebo) separately (Figure S1), and for LSD and PSB (Figure S6). This essentially implies that none of the findings are specific to the difference between ketamine and placebo, but instead reflect a general feature of the methodology used here. Please report correlations between these maps or provide some other demonstration that the reported findings are indeed specific to the ketamine-placebo contrast.

We thank the reviewer for highlighting this important issue. To address this, we first z scored the PC1 maps for placebo, ketamine, ∆ ketamine, ∆ psilocybin, ∆ LSD and presented them on the same scale and direction (given the direction of a principal component is inherently arbitrary) in order to be able to directly qualitatively compare the maps (Figure 1—figure supplement 7A-E). We then calculated the correlations between PC1-5 for the placebo, ketamine, ∆ ketamine, ∆ psilocybin, and ∆ LSD data (Figure 1—figure supplement 7F). For PC1, we see that as expected the highest correlations (*r* > 0.8) are between ∆ ketamine PC1 and placebo PC1, and ∆ ketamine PC1 and ketamine PC1 as these involve the same sample of participants (Figure 1—figure supplement 7F). In contrast, the PC1 correlations between ∆ ketamine PC1 and ∆ psilocybin PC1 /∆ LSD PC1 are lower (*r* ≈ 0.5), indicating the maps are not identical. Furthermore, the moderate correlations between ∆ ketamine PC1 and ∆ psilocybin PC1/∆ LSD PC1 are not entirely surprising given that there is overlap in the subjective effects of these highly psychoactive substances. In contrast, ∆ ketamine PC2-5 are more dissimilar to ∆ psilocybin/LSD PC2-5: PC2 *r* = 0.26/0.22, PC3 *r* = 0.18/0.05, PC4 *r* = 0.1/0.02, and PC5 *r* = -0.11/-0.02 (Figure 1—figure supplement 7F). While these results do not rule out the possibility that PC1 does to some extent capture a general feature of the methodology, we believe that in particular the moderate correlations between ∆ ketamine PC1 and ∆ psilocybin PC1 /∆ LSD PC1 demonstrate that the findings do show specificity to the ketamine-placebo contrast. Although a thorough comparison of the PCA results for ∆ ketamine, ∆ psilocybin, and ∆ LSD is beyond the scope of this manuscript (see reply to comment 2 for more details on how this will be investigated in the future), to address this issue in the manuscript we have now: (1) z-scored all psilocybin and LSD ∆ GBC maps to enable a reader to directly compare them with the ketamine ∆ GBC maps (see Figure 1—figure supplement 7); and (2) included the PC2 maps for ∆ psilocybin and ∆ LSD (see Figure 1—figure supplement 6).

– LSD and PSB are only used for evaluating dimensionality. A more convincing analysis would compare the primary effects across all drugs to identify the specific effects of ketamine.

We agree with this feedback and are working on a collaborative analysis with another team (Prof. Robin Carhart-Harris) where we have pooled all our effects into one mega-analysis to evaluate precisely this question with sufficient power, after the observation in the present manuscript. We agree that a full analysis of the ketamine effects to other pharmacological manipulations is necessary and would be convincing. We now discuss this point more extensively in the Discussion section to highlight the importance of testing this hypothesis. That said, we hope the Reviewer and the Editor can appreciate that a full analytic expansion to independent pharmacological datasets would be beyond the scope of the present manuscript. The present manuscript, even in its current form, is quite extensive; therefore, while we completely embrace the thesis that generalizing the analytic approach and evaluating the neuro-behavioral effects in each pharmacological intervention is vital, it is our strong preference to focus on ketamine in this paper because of its specific mechanism of action, which motivated some of the strong a-priori cortical gene expression analyses. Collectively, we sincerely hope that a discussion of this analytic direction (which is actively in progress) addresses the core notion that applying these methods in other pharmacological manipulations would be impactful. We have now added this point as an additional future direction to the Discussion section:

“Given the differences in effective dimensionality we see across different drugs, an important direction for future research will be an expansion of the primary analysis to LSD and psilocybin in order to more thoroughly compare their effects.”

– There is no assessment of the specificity of the gene expression correlations. Many different genes show similar spatial patterns. In fact, the PC maps are patterns along the classical sensory fugal hierarchy, which some of these authors have shown to be the dominant PC of gene expression (Burt, Nat Neuro), meaning that many different genes will have similar expression profiles. Please provide stronger evidence supporting the specificity of the effects on the selected genes with respect to appropriate null models (e.g., Wei et al., Hum Brain Mapp, 2022). This is critical to support the main claims associated with this analysis. This analysis should be accompanied by a more detailed explanation of why the specific genes were chosen, as many other choices are possible.

We absolutely agree that as many different genes show similar spatial patterns our analysis does not provide an assessment of specificity; however, for this reason we specifically picked two genes that are associated with interneuron classes that have been shown in the animal literature to play a mechanistic role in ketamine’s effects. Precisely to the Reviewer’s point, because a vast majority of genes will express along the sensoryfugal axes, we picked two genes that are a priori mechanistically implicated in ketamine’s effects on the brain. Specifically, PVALB (which encodes parvalbumin interneurons), and SST (which encodes somatostatin interneurons), may both may be preferentially antagonized by ketamine. The point of the analysis is not to overstate that it is specifically PVALB and SST, as opposed to any other gene that follows a similar expression pattern, that are related to ketamine’s effects, but rather to test the hypothesis that the low rank solution of the pharmacological effects, which produces a dominant axis of variation, is more strongly associated with the SST/PVALB cortical gene expression maps compared to the mean-effect of ketamine. We now provide a more nuanced interpretation of the results in the Discussion section, and have updated the introduction outlining more clearly why we chose PVALB and SST, as well as explicitly stating the limitations of the conclusions that can be drawn from this analysis, and directly referencing the Burt, Nat Neuro paper highlighted by the reviewer:

“In order to assess the relative merits of characterizing ketamine’s acute effects using using a uni-dimensional (i.e. mean-level) or multi-dimensional (i.e. PCA) approach, will compare which results best relate to ketamine’s hypothesized molecular mechanisms. We will focus on the indirect hypothesis, which posits that ketamine first inhibits tonic-firing GABAergic interneurons via NMDAR blockade, which in turn leads to a burst of glutamate that drives synaptic plasticity ((11)). Specifically, we will test the hypothesis that individual differences in ketamine’s neural systems-level effects are associated with SST and PVALB GABAergic interneurons by correlating ketamine’s neural effects with SST and PVALB cortical gene expression maps derived from the Allen Human Brain Atlas (AHBA). SST and PVALB genes were selected a-priori based on: (i) the large body of animal literature directly linking SST and PVALB interneurons to ketamine’s acute effects ((11, 22, 26)); (ii) studies showing SST and PVALB interneurons are particularly sensitive to ketamine as a result of their NMDA receptor subunit configuration ((25)); (iii) previous research showing that specific cortical regions express different ratios of SST and PVALB interneurons, with SST showing higher expression levels in association relative to sensory regions, while PVALB shows the reverse pattern ((1, 14)); and (iv) studies of patients with depression have reported reduction in both SST or PVALB interneurons in areas such as the hippocampus and PFC ((7, 8, 24, 32)). Given that many different genes show similar spatial patterns in the brain, with the most common pattern explaining approximately 25% of the total gene expression variance ((4)), it is important to note that by relating ketamine’s neural effects to SST and PVALB cortical gene expression patterns we do not make any claims regarding gene specificity. Instead, we use the established relationship between ketamine and SST/PVALB interneurons evaluate the relative merits of assessing ketamine’s acute effects using a multi-dimensional vs. a uni-dimensional approach.”

– The discrepancy between mean and PC approaches is expected if a significant variance is explained by >1 PC, so this is not particularly surprising. The rationale for the emphasis on this result is unclear

We thank the reviewer for raising this point which is of central importance to the paper. Although the discrepancy between the mean and PC approach is certainly expected when a significant variance is explained by >1 PC, the use of a principal component approach is still not very common when analysing ketamine’s effects. Most studies still use a mean difference approach, which obscures the possibility that there are multiple axes of ketmaine response. To make this more clear in the manuscript, we have now rewritten the introduction with a greater focus on the motivation of the paper and how the PCA fits into this. Of particular relevance are the first and fourth paragraphs, also included below:

Paragraph 1: “Over the last two decades, ketamine has emerged as one of the most promising therapies for treatment-resistant depression (TRD) ((19)). Robust individual differences in response to ketamine have been observed in both healthy controls and patients with Treatment Resistant Depression (TRD) (12, 21). For example, a single ketamine infusion results in a response rate of around 65% in patients with TRD, while individual differences in baseline molecular effects (e.g. NMDA receptor occupancy) and brain function predict the degree to which an individual experiences specific acute ketamine-induced symptoms ((16, 21, 29)). Despite these findings, an assumption persists within the fMRI pharmacology literature that the behavioral and neural effects of substances like ketamine are uniform across individuals, and that the central tendency can effectively capture these effects. In this study we challenge this assumption and instead posit that ketamine’s effects are multi-dimensional, and that these dimensions will capture individual differences in response to ketamine.”

Paragraph 4: “The main limitation of relying on the power of central tendency is that meaningful differences between individual subjects may be lost through the process of averaging. One method that has been used to successfully address this and uncover individual differences in psychiatric research is a principal component analysis (PCA) ((13, 18)). A PCA is a data-driven method that is able to uncover both group-level and individual-level differences. More specifically, a PCA allows us to test the hypothesis that ketamine’s effects are multi-dimensional: if ketamine’s effects are uniform then we would expect a PCA to result in one principal component on which all the participants can be mapped, however if there are systemic differences between participants we would anticipate a PCA to generate multiple principle components. A PCA is also able to capture individual-level differences through an individual’s relative positioning along the axes of the principal components ((13)).”

– The analysis of high and low loaders, which occupies significant volumes of text, is highly qualitative. Please provide full distributions of relevant metrics across all participants and statistical analysis of the entire sample.

We agree with the reviewer that this analysis should be taken further as we make a qualitative claim when it is possible to determine the relationship quantitatively. To address this, we have now removed Figure S8 in the original manuscript and updated the analysis in Figure 5 by removing the reference to high and low loaders and replacing it with a new analysis that allows us to explore individual variation in relation to the neural-behavioral mapping in a quantitative manner. As suggested by the reviewer, instead of examining the relationship between ∆ GBC and SST/PVALB cortical gene expression maps only for the highest and lowest loaders, we now do so for the entire sample and make inferences only on the statistical analysis of the combined results of the entire sample. As a brief overview, we now: (i) compute the relationship between each individual subject’s ∆ GBC and SST/PVALB cortical gene expression maps (see four additional supplemental figures Figure 5—figure supplement 1-4); (ii) order participants in terms of their neuro-behavioral PC1 score (calculated by correlating their ∆ GBC map with their neural-behavioral PC1 map) and plot each participant’s observed r-values to provide an overview of the results in a single summary figure (Figure 5G); and (iii) quantitatively establish if there is relationship between distinct SST and PVALB gene expression map associations and the neuro-behavioral PC1 scores across the entire sample (Figure 5H). This is outlined fully in the updated results subsection ‘Neuro-behavioral mapping captures individual variation in how SST and PVALB interneurons may be differentially impacted by ketamine’. We are extremely grateful to the reviewer for this suggestion and believe the new analysis greatly strengthens the paper. Finally, we also updated the Discussion section as follows:

“Furthermore, we demonstrated that neuro-behavioral PC1 captured individual variation in how SST and PVALB interneurons may be differentially impacted by ketamine: we found an association between participants’ neuro-behavioral PC1 scores and whether their ∆ GBC maps positively correlated with SST gene expression maps and negatively correlated with PVALB gene expression maps (seen in 58% of participants), or whether they showed the reverse pattern (seen in 12% of participants) (Figure 5H).”

– Many different analyses are presented and the figure captions are dense. The readability of the paper could be improved if the results were more streamlined, particularly since it is based on a small number of participants. Some suggestions:– Remove comparisons with mean GBC and focus on the PCs;– Focus on the neuro-behavioral results and genetic mapping.

While we certainly appreciate the Reviewer’s direct suggestions, we hope that we can articulate why keeping the comparison between the mean GBC and PCs is essential. First, the mean effect is showing what prior literature has produced, and is vital to anchor the reader’s perspective to what the result would be under the null hypothesis that there is a single dimension under ketamine response (in which case the mean would be a valid measure). As a direct corollary, removing the neural PCA would fundamentally change the scope of the paper and remove what we believe is the core finding of the paper, namely that ketamine induces a neural response pattern that varies across individuals in a multi dimensional manner. Again, we want to underscore this point, because the current framework in the vast majority of pharmacological imaging studies (including our own prior work) has made the assumption that if a pharmacological agent is administered to a sample of healthy adults, the response will be along a uniform neural circuit that exhibits variation along an axis of response. The alternative hypothesis is that there are quantifiable functional differences between individuals in the pattern of their neural response to a given pharmacological agent. Testing this hypothesis, both neurally and behaviorally, is the essence of the study, because it rejects the hypothesis that the mean is the best measure of ketamine’s acute effects. We hope that this response removes any concerns regarding the (in our mind) vital findings. Finally, while we appreciate the reviewer’s concerns that N=40 may be small,

to our knowledge this is the largest single-site study of ketamine’s neural and behavioral patterns while in scanner. In addition, we have now re-written the introduction to more clearly outline the motivation for the paper, and hope that it is now also clear in the manuscript from the outset why the comparison between the mean GBC and PCs is crucial to the paper. For example, see the updated first paragraph of the introduction:

“Over the last two decades, ketamine has emerged as one of the most promising therapies for treatment-resistant depression (TRD) ((19)). Robust individual differences in response to ketamine have been observed in both healthy controls and patients with Treatment Resistant Depression (TRD) ((12, 21)). For example, a single ketamine infusion results in a response rate of around 65% in patients with TRD, while individual differences in baseline molecular effects (e.g. NMDA receptor occupancy) and brain function predict the degree to which an individual experiences specific acute ketamine-induced symptoms ((16, 21, 29)). Despite these findings, an assumption persists within the fMRI pharmacology literature that the behavioral and neural effects of substances like ketamine are uniform across individuals, and that the central tendency can effectively capture these effects. In this study we challenge this assumption and instead posit that ketamine’s effects are multi-dimensional, and that these dimensions will capture individual differences in response to ketamine.”

– It would be helpful to see the distribution of the ketamine response on the behavioral variables, frequently the range of this response is small in healthy volunteers and makes analyses difficult. Are there participants who don't have a significant response or outliers?

We thank the reviewer for raising this point and agree it is important to show that the ketamine response as measured via the behavioral variables has sufficient range. To address this we now include two supplementary tables showing: (i) only 5/40 participants do not show a significant PANSS response; (ii) outliers are limited with 10 behavioral items containing one outlier, 1 behavioral item showing 2 outliers, and 2 behavioral items showing 3 outliers (Table S4, Table S5). We have also included this additional sentence in the methods outlining this:

“Behavioral response to ketamine assessed via PANSS showed sufficient range to justify further analysis, as 35/40 participants showed a significant difference in PANSS following ketamine, while outliers were limited (with the exception of items in which the interquartile range was 0) (Table S4, Table S5).”

– Were the physiological measures recorded during or after the infusions? Typically heart rate and blood pressure increase during acute ketamine administration and the same is likely true for acute LSD or psilocybin. Please provide an indication of how these changes affect the GBC

This is an excellent point and something we will certainly consider in future studies. While physiological measures were collected in the ketamine study to ensure participants remained within a certain safe range, the data was not analysed. However, as the reviewer suggested there is evidence that at these doses all three drugs increase blood pressure and/or heart rate. In addition, one study directly comparing LSD and psilocybin indicated that they have differential effects on blood pressure and heart rate, with psilocybin having greater impact on blood pressure and LSD on heart rate. We have now added this as a discussion point:

“Another limitation is that we did not regress out physiological measures. There is a large body of evidence to suggest that heart rate and blood pressure impact resting-state functional connectivity, and sub-anesthetic doses of ketamine and moderate doses of psilocybin and LSD have all been shown to increase heart rate and blood pressure ((3, 5, 6, 17, 23, 28)). Furthermore, while to date no studies have directly compared the physiological effects of all three drugs, there is evidence that LSD and psilocybin have distinct effects on blood pressure and heart rate ((15)).”

– Please clarify how double-blinding was implemented precisely.

We are extremely grateful to the reviewer for catching this mistake: the ketamine study was single-blinded, with participants blinded to their treatment condition, while the LSD and psilocybin studies were doubleblinded. We have now corrected this mistake in the manuscript.

– There might be other sources for inter-individual variation of ketamine's response such as use-dependent excitation of NMDA networks, sleep profiles or genetic variation of NMDA receptor subunits. The authors may want to incorporate these into their explanation of inter-individual variability.

We extremely grateful to the reviewer for these invaluable suggestions regarding additional sources for the inter-individual variation in response to ketamine and have incorporated them in the Discussion section as follows:

“Meanwhile, the bi-directional nature of the principal components may relate to individual differences in a number of factors including use-dependent excitation of NMDA networks, genetic variation in NMDA receptor subunits, and sleep profiles (9, 10, 20, 31).”

– Please explain in more detail why the PANSS or δ-PANSS (Post-Pre) was used for the PCA. Why not CADSS, which seemed to have stronger effects than the PANSS (Table S1)?

We thank the reviewer for highlighting the need for clarity regarding this point. We focused on PANSS and cognition rather than CADSS and cognition in order to create a more parsimonious model that we can relate to prior work that has been conducted in the field of psychosis using the same analysis approach (i.e. running a PCA on PANSS and cognition data and regressing it onto parcellated neural GBC data – see Ji et al. 2020) ((18)). We also wanted to be able to compare our findings more generally to the schizophrenia literature, and PANSS is a typical measure in the psychopathology literature that has been used as a FDA regulated endpoint for psychosis treatments. We chose not to include both PANSS and CADSS as this solution resulted in the amount of total variance explained going down from 41.1% to 29.9% (Figure 3—figure supplement 2).

However, we agree with the reviewer that capturing the full spectrum of pharmacologically induced behavioral effects is vital, and that focusing on CADSS and cognition may be an equally valid approach given the PCA run on CADSS and cognition results in a total variance explained of 51.5% (Figure 3—figure supplement 3). Furthermore, the CADSS and cognition PCA solution is very different to the PANSS and cognition PCA solution: four vs. two principal components are identified, and when we regress the principal components onto the parcellated neural GBC data we see very little overlap in the PANSS and cognition vs. CADSS and cognition neuro-behavioral PC maps (max r = 0.38) (Figure 4—figure supplement 1). Overall, this indicates that a comparison of the relative merits of CADSS vs. PANSS in relation to this analysis strategy may be an interesting avenue for future research, although it is beyond the scope of this paper. However, to highlight this point to the reader we now include three additional figures showing: (i) PCA results for PANSS, CADSS and cognition; (ii) PCA results for CADSS and cognition; (iii) a comparison of the neuro-behavioral maps from the three PCA solutions (PANSS and cognition; PANSS, CADSS and cognition; CADSS and cognition) (Figure 3—figure supplement 2-3, Figure 4—figure supplement 1). We also added this sentence to the methods section:

“We did not include CADSS in the model as doing so resulted in the total amount of variance explained by the PCA going down from 41.1% to 29.9% (Figure 3—figure supplement 2). For comparison we ran a PCA on (i) PANSS, CADSS, and cognition (Figure 3—figure supplement 2) and (ii) CADSS, and cognition (Figure 3—figure supplement 3), and directly compared the different PCA versions by correlating the resulting neuro-behavioral maps (Figure 4—figure supplement 1).”

– Please discuss the downsides of using repetitive rating scales in such an endeavor. To this end, I would be interested in the author's view on what explains more variance in neuropsychological effects of repetitive ketamine/lsd/psilocybin experiments: visual analog scales, questionnaires, or tasks.

We thank the reviewer for highlighting this important broader limitation and agree that the use of repetitive rating scales when assessing the neuropsychological effects of ketamine, LSD, and psilocybin is not without its downsides. For example, the frequency and timing of repetitive rating scales may introduce order effects and participant fatigue, affecting the quality of the data. In addition, the repetitive rating scale used in this study, PANSS, relies of self-report data, making it subject to biases that may reflect the validity of the results. For example, Likert-type scales are often associated with a response bias where participants tend to select responses in a pattern (30). However, the use of repetitive self-report and clinician-rated scales to capture how a pharmacological manipulation alters neuropsychological symptoms is standard practice in psycho-pharmacological experimental as well as FDA regulated clinical trials, and we are not currently aware of another option with which to evaluate whether a drug has an effect on symptoms without measuring those symptoms before and after the administration of a drug.

While other measures such as tasks and visual analog scales do mitigate some of these limitations, as tasks provide a more objective measure while visual analog scales may capture more subtle differences in the subjective experience and reduce the influence of response bias associated with Likert-type scales, they would still need to be assessed at multiple timepoints. Therefore many of the same limitations would apply e.g. a task would be subject to practise effects. Ultimately, an approach that combines multiple assessment tools, integrating both objective and subjective measures, is preferable to provide a comprehensive overview of a substance’s neuropsychological effects. In this study we do combine a task (Spatial Working Memory Task), that objectively captures a specific cognitive effect, and a subjective measure (PANSS), that aims to capture the broader psychological state and experience. However, the addition of other measures such as visual analog scales could certainly be advantageous for future studies depending on the specific research question.

Finally, in terms of what explains more variance in the neuropsychological effects of repetitive ketamine, LSD, and psilocybin experiments, to date we are unaware of any previous research that has conducted a systemic investigation directly comparing the predictive power of visual analog scales, questionnaires, and cognitive tasks within the same experiment. However, in this ketamine dataset, we certainly see that both the task and questionnaire measures contribute to the behavioral PCs.

In summary, we appreciate the importance of the reviewer’s suggestion that practice and familiarity effects are an important potential psychometric confound and have added an additional sentence in the limitation section outlining this:

While behavioral measures were collected during both peak effects and offset, the use of repetitive question-naires and tasks introduces potential psychometric confounds such as familiarity and practice effects.

– How were sex and age effects addressed in the main results?

We thank the reviewer for raising this excellent point and have now added an additional supplemental figure showing there is no correlation between ∆ PC1-5 scores or behavioral PC1-2 scores and age (Figure 6—figure supplement 5C,F,I,L,O,R,U). In addition, we found no significant difference in ∆ PC1-2,4-5 scores or behavioral PC1-2 scores between men and women (Figure 6—figure supplement 5B,E,K,N,Q,T). While we do find a significant difference between ∆ PC3 scores (*p-uncorrected* = 0.03), further exploration of PC3 is beyond the scope of this manuscript (Figure 6—figure supplement 5H). However, the possibility that one PC may be related to differences in sex would certainly be an extremely interesting avenue for future research, especially if we are able to replicate this finding with a larger sample and we have now added this point to the Discussion section:

“To unpack the variation in PC3-5 ∆ GBC a larger sample size is needed, especially given each PC represents a bidirectional axis of variation. A larger sample size would also allow us to investigate alternative models for ketamine’s molecular mechanisms in addition to the indirect hypothesis, such as how ketamine’s effects differ based on NMAR subtypes, or excitatory vs. inhibitory synapses. It would also enable us to explore whether any demographic characterises such as sex relate to specific PCs.”

– Effective dimensionality was calculated using the participation ratio and dataset re-sampling was used to control for sample size in this calculation, but dimensionality is also affected by motion within the sample among other noise sources, which are not well discussed. In particular, each drug may affect physiological noise in different ways and this may in turn affect their dimensionality measurement. A more detailed comparison across conditions of the effects of motion and the global signal is required. For instance, the authors could compare FD statistics and properties of the global signal (e.g., variance) and physiological recordings.

We agree with the reviewer that dimensionality may also be affected by motion within the sample, and that a more detailed comparison across the different studies (ketamine, psilocybin, and LSD) is warranted. We have now included an additional supplemental figure examining the relationship between motion and global signal variance across the three studies. Specifically, we show that there is no relationship between mean FD and mean GBC variance across all participants for the placebo or substance scan sessions in the ketamine study or the psilocybin study (Figure 1—figure supplement 5A-B). While we do find a significant correlation between mean FD and mean GBC variance for the substance scan session in the LSD study (r=0.52, p=0.01), this correlation is driven by a single outlier which when removed results in a non-significant correlation (r=-0.05, p=0.83) (Figure 1—figure supplement 5C). Furthermore, when we compared the correlations between mean FD and mean GBC across all three studies using Fisher-Z-Transformation, we found no difference in correlations between any of the studies (even with the LSD outlier left in). Finally, when we directly compared FD in each of the three studies, we found that while FD in both the placebo and substance sessions was significantly higher in the ketamine study compared to the LSD and psilocybin study, we found no difference in ∆ (substance – placebo) FD between the three studies (and it is ∆ (substance – placebo) that is the input for all the analyses included in the manuscript) Figure 1—figure supplement 5E.

Given we see ketamine shows higher effective dimensionality compared to LSD and psilocybin, and ketamine shows increased FD compared to LSD and psilocybin in the placebo and substance scan sessions, we wanted to investigate whether ketamine’s higher FD could be related to its higher effective dimensionality score. To explore this, we separated the ketamine study participants into those with the higher FD values (i.e. the ’high motion’ group) and those with the lower FD values (i.e. the ’low motion’ group). We then calculated effective dimensionality for the high and low motion groups, and found that effective dimensionality was significantly higher in the low motion group compared to the high motion group for the placebo, substance, and ∆ scan sessions Figure 1—figure supplement 5F. This indicates that the higher motion seen in the ketamine study is not what is driving its higher effective dimensionality score as here we see the opposite pattern: effective dimensionality is higher in the low motion group. Overall, while we cannot rule out that motion may have some influence on effective dimensionality scores, we do not find any evidence to indicate that motion is driving the differences in effective dimensionality between the studies as: (i) we do not find any relationships between mean FD and GBC variance; (ii) there is no difference in ∆ (substance – placebo) FD between the three studies; and (iii) when we compare high motion and low motion participants in the ketamine group we do not find that higher motion is related to higher dimensionality (Figure 1—figure supplement 5).

– On a related point, the description of the FD cutoff for scrubbing frames requires more detail. Is the median taken across all scans and all participants? Or all scans within each participant separately? What was the final cutoff determined by this procedure?

After the HCP minimal preprocessing pipelines, movement scrubbing was performed ((27)). Bad frames with possible movement-induced artifactual fluctuations in intensity were identified if they met at least one of the following two criteria ((2)). First, framewise displacement (FD) was computed by summing the displacement across all six rigid body movement correction parameters. Frames in which FD exceeded 0.5mm were flagged. Secondly, normalized root mean square (RMS) was calculated by taking the root mean square of differences in intensity between the current and preceding frame across all voxels and dividing it by the mean intensity. Frames in which RMS exceeded 1.6x the median across the scans (calculated separately for each participant) were also flagged. The flagged frames, as well as the frame immediately preceding and immediately following any flagged frames, were discarded from further analyses. Subjects with more than 50% flagged frames were excluded completely.